# Differentially Private Equilibrium Finding in Polymatrix Games

**Mingyang Liu, Gabriele Farina & Asuman Ozdaglar**
LIDS, EECS
Massachusetts Institute of Technology
Cambridge, MA 02139, USA
`{liumy19,gfarina,asuman}@mit.edu`

## Abstract

We study equilibrium finding in polymatrix games under differential privacy constraints. Prior work in this area fails to achieve both high-accuracy equilibria and a low privacy budget. To better understand the fundamental limitations of differential privacy in games, we show hardness results establishing that no algorithm can simultaneously obtain high accuracy and a vanishing privacy budget as the number of players tends to infinity. This impossibility holds in two regimes: (i) We seek to establish equilibrium approximation guarantees in terms of Euclidean *distance* to the equilibrium set, and (ii) The adversary has access to all communication channels. We then consider the more realistic setting in which the adversary can access only a bounded number of channels and propose a new distributed algorithm that: recovers strategies with simultaneously vanishing *Nash gap* (in expected utility, also referred to as *exploitability*) and *privacy budget* as the number of players increases. Our approach leverages structural properties of polymatrix games. To our knowledge, this is the first paper that can achieve this in equilibrium computation. Finally, we also provide numerical results to justify our algorithm.

## 1 Introduction

Many multi-agent settings are challenging because each participant's action can influence all others. However, in many settings of interest, the interaction is more localized and tractable. Polymatrix games provide a tractable and rich model that can capture such settings (Janovskaja, 1968; Cai et al., 2016; Deligkas et al., 2017). In polymatrix games, players engage in pairwise interactions defined by an underlying graph in which nodes represent players and edges model the interaction between the two players connected by an edge. Each player chooses the same strategy in each pairwise game she is involved with, and her utility is the sum of her utilities for each of these games.

In several settings, such as security games (De Nittis et al., 2018) and financial markets (Evangelista et al., 2022; Donmez et al., 2024), the players' utility functions may be sensitive and need to be kept private by the players as they update their behavior toward equilibrium. Consider, for instance, a decentralized bilateral trading network. Here, players are traders, actions represent pricing strategies, and graph edges denote trade relationships. While players aim to reach a market equilibrium to maximize profit, their specific utility matrices, which encode sensitive reservation prices or private valuations, must remain confidential to prevent exploitation in future bargaining.

Typically, there are two approaches for finding equilibrium while keeping the utility functions private: centralized and distributed methods. In centralized methods, the players send their utility functions to a trusted central server (Kearns et al., 2014; Rogers & Roth, 2014; Cummings et al., 2015). When a trusted central server is infeasible, the only viable option for computing equilibrium is for the players to use a distributed algorithm where they make local computations and exchange information with their neighbors. Throughout the process, a malicious adversary may inspect the communication channel and infer the utility function, which might reveal sensitive information about individual preferences. To keep the computation secure and prevent the leakage of sensitive information, tools from differential privacy (DP) (Dwork et al., 2006) can be employed.

Several authors have considered the question of differentially private equilibrium computation in games. However, previous work was either unable to achieve high accuracy and low differential privacy budget simultaneously (Ye et al., 2021; Wang et al., 2022), or only achieved a weaker form of differential privacy where the adversary is still able to infer part of the private information (Wang & Başar, 2024; Wang & Nedić, 2024). We defer a broader discussion on prior work to Appendix A.

## CONTRIBUTIONS AND TECHNIQUES

In this paper, we study the problem of distributed, differentially private equilibrium computation in polymatrix games. We build on the notion of adjacent distributed optimization problem defined in Huang et al. (2015). In particular, given a polymatrix game $\mathcal{G}$ we define *adjacent games* as those that differ from $\mathcal{G}$ on the utility function of a single player (see Figure 1 (a) and Definition 2.1). Then, once the algorithm for equilibrium finding is differentially private, an adversary cannot distinguish, with high probability from his observations, the original game from any other games that are adjacent to it. Hence, when differential privacy holds, the adversary is unable to determine the utility function of any player with high confidence, ensuring the equilibrium computation process is privacy-preserving.

**Impossibility.**    Using this concept of adjacency, we begin by showing that finding an approximate equilibrium with arbitrarily small differential privacy guarantees can be impossible, depending on the desiderata imposed on the process. Specifically, we show that if the adversary can access an arbitrary number of communication channels, or the target accuracy metric of interest is the Euclidean distance from the equilibrium set, then distributed computation of high-quality equilibria while providing vanishing privacy guarantees is impossible. In other words, one will inevitably suffer either a low approximation of the equilibrium, or a low level of privacy (high privacy budget). While these negative results serve as guardrails to guide algorithm design, it is important to realize that they do not preclude all paths to meaningful results. Indeed, it is important to realize that Euclidean distance to the equilibrium set is only one of the set of possible quality metrics for measuring equilibrium approximation. Another metric is *Nash gap* or *exploitability*, meaning how close a strategy is to equilibrium in terms of *expected utility* rather than metric distance to the equilibrium set in strategy space. This is particularly relevant in practice, where potential gains from deviation are more important than proximity in strategy space.

**Positive Results.**    Finally, we complement our negative result with a positive result regarding exploitability guarantees, corroborated by experiments in Appendix E. In particular, we propose a new distributed algorithm for computing a coarse correlated equilibrium (see Figure 1 (b)). As is typical in the differential privacy literature, our algorithm involves each player communicating a noisy version of their strategy to their neighbors and updating their strategy using a regularized proximal gradient step. A key novelty of the algorithm is to scale the regularizer proportional to the harmonic mean of the degrees divided by the degree of the player. The inverse proportionality with the degree ensures that more regularization is introduced for players with a lower degree (since a low-degree player's gradient is more sensitive to variations in the utility matrices of its neighbors).

**High-Level Intuition.**    We propose Algorithm 1, which simultaneously achieves low exploitability and low DP budget, with guarantees that improve as the number of players $N$ increases. Moreover, the algorithm achieves that whenever the associated graph with the polymatrix game is sparse or dense, by imposing an adaptive regularizer on players' utility functions. When the graph associated with the polymatrix game is sparse, we leverage the fact that for most nodes that are at least of distance $t$ to the edge of the changed utility matrices, changes in the utility matrices between adjacent polymatrix games have minimal impact on them during the first $t$ updates. In contrast, when the graph is dense, additional regularization, which is inversely proportional to the players' degrees, stabilizes the updates of players with low degrees so that the adversary's observation in two adjacent games will be similar. For players with high degrees, the aggregative nature of the utility function in polymatrix games mitigates the effect of changes in the utility matrices.

## 2    PRELIMINARIES

We now review several key concepts related to differential privacy and polymatrix games.

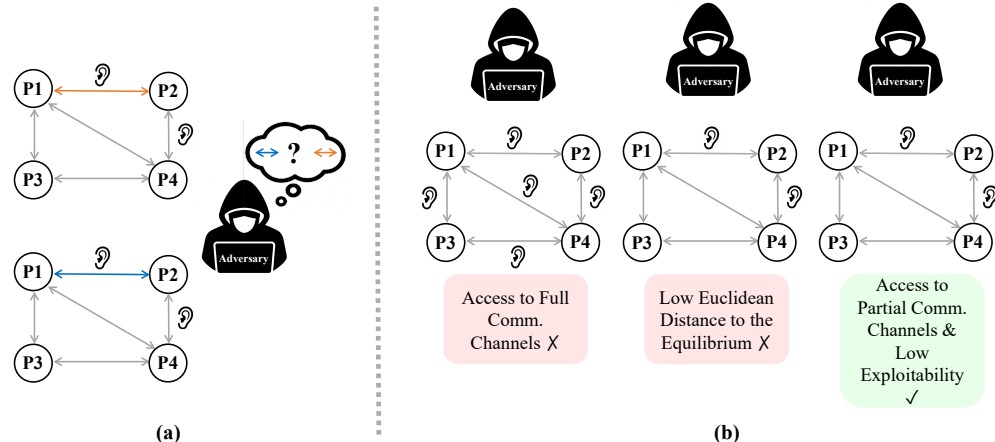

**(a)**                                         **(b)**

Figure 1 (a): An illustration of adjacent polymatrix games defined in Definition 2.1. The nodes represent the players, and the lines represent the edges of the polymatrix game. The two games differ at the blue/orange edge, and the adversary cannot differentiate these two games with certainty from his observations. 🎧 on an edge means that the communication channels between those two players can be accessed by the adversary.

Figure 1 (b): The illustration of results: the impossibility when the adversary can access all communication channels showed in Lemma 3.1 (left), the impossibility to achieve low Euclidean distance showed in Lemma 3.2 (middle), and the positive results showed in Theorem 7.1 (right).

**Basics.** For any integer $N > 0$, let $[N] := \{1, 2, 3, \cdots, N-1, N\}$. For any vector $\boldsymbol{x}$, we use $\|\boldsymbol{x}\|_p$ to denote its $p$-norm. By default, $\|\boldsymbol{x}\|$ denotes the Euclidean norm $\|\boldsymbol{x}\|_2$. The symbol $\Delta^n := \{\boldsymbol{x} \in [0,1]^n \colon \sum_{i=1}^n x_i = 1\}$ denotes the $(n-1)$ dimensional probability simplex. Moreover, for any discrete set $\mathcal{S}$, we can define $\Delta^{\mathcal{S}}$ as the probability simplex over $\mathcal{S}$, with each index as the element in $\mathcal{S}$. Hence, $\Delta^{[N]} = \Delta^N$. For any convex set $\mathcal{C}$, we use $\operatorname{Proj}_{\mathcal{C}}(\boldsymbol{x}') := \operatorname{argmin}_{\boldsymbol{x} \in \mathcal{C}} \|\boldsymbol{x} - \boldsymbol{x}'\|^2$ to denote the projection to $\mathcal{C}$ from a vector $\boldsymbol{x}'$ with respect to Euclidean distance.

**Polymatrix Games.** Polymatrix games can be written as a tuple

$$\mathcal{G} := \Big( [N], E, \{\mathcal{A}_i\}_{i \in [N]}, \{\boldsymbol{U}_{i,j}\}_{(i,j) \in E} \Big),$$

where

- $[N]$ is the set of players.
- $E$ is the set of edges, where each $(i, j) \in E$ indicates that players $i, j$ interact with each other.
- $\{\mathcal{A}_i\}_{i \in [N]}$ is the set of players' action set, which means player $i \in [N]$ chooses actions in $\mathcal{A}_i$.
- Let $A := \max_{i \in [N]} |\mathcal{A}_i|$ be the size of the largest action set.
- $\boldsymbol{U}_{i,j} \in [-1, 1]^{\mathcal{A}_i \times \mathcal{A}_j}$ is the utility matrix between player $(i, j) \in E$.

For each player $i \in [N]$, we denote with the symbol $\mathcal{N}(i) := \{j \in [N] \colon (i, j) \in E\}$ the set of her neighbors. Then, when each player $k \in [N]$ chooses her strategy as $\pi_k \in \Delta^{\mathcal{A}_k}$, the utility of player $i$ is $\frac{1}{|\mathcal{N}(i)|} \sum_{j \in \mathcal{N}(i)} \pi_i^\top \boldsymbol{U}_{i,j} \pi_j$ [1]. Moreover, when the strategy profile of all players is $\boldsymbol{\pi} := (\pi_1, \pi_2, \cdots, \pi_N)$, we can define the gradient of player $i$'s strategy with respect to her loss function (the negative of utility function) as $\boldsymbol{g}_i^{\boldsymbol{\pi}} := -\frac{1}{|\mathcal{N}(i)|} \sum_{j \in \mathcal{N}(i)} \boldsymbol{U}_{i,j} \pi_j$.

---

[1] Typically the definition of player $i$'s utility is $\sum_{j \in \mathcal{N}(i)} \pi_i^\top \boldsymbol{U}_{i,j} \pi_j$. However, to ensure that the function and its gradient are bounded by constants regardless of the game size, we divide it by $|\mathcal{N}(i)|$, which aligns with the DP literature (Huang et al., 2015; Wei et al., 2020; Ye et al., 2021). Note that such modification will not change the equilibrium.

**Differential Privacy.** We adopt the notion of adjacency in Huang et al. (2015). For any two polymatrix games, they are *adjacent* to each other if and only if they only differ from the utility matrices on a single edge. An intuitive illustration can be found in Figure 1 (a). Formally,

**Definition 2.1** (Game adjacency). Given two polymatrix games $\mathcal{G} = \left([N], E, \{\mathcal{A}_i\}_{i \in [N]}, \{\boldsymbol{U}_{i,j}\}_{(i,j) \in E}\right)$ and $\mathcal{G}' = \left([N'], E', \{\mathcal{A}'_i\}_{i \in [N']}, \{\boldsymbol{U}'_{i,j}\}_{(i,j) \in E'}\right)$, they are said to be *adjacent*, indicated as $\mathcal{G} \sim \mathcal{G}'$, if

1. $N = N'$, $E = E'$ and $\mathcal{A}_i = \mathcal{A}'_i$ for any $i \in [N]$; and
2. except for an edge $(i, j) \in E$, $\boldsymbol{U}_{i',j'} = \boldsymbol{U}'_{i',j'}$ and $\boldsymbol{U}_{j',i'} = \boldsymbol{U}'_{j',i'}$ for any $(i', j') \in E \setminus \{(i, j), (j, i)\}$.

Consider now a generic iterative algorithm for finding equilibrium in polymatrix games, and denote with $\Theta$ the space of all possible internal states of the algorithm at any time. Then, for a given polymatrix game $\mathcal{G}$, when the algorithm runs $t > 0$ timesteps, it will generate intermediate variables $\boldsymbol{\theta} := \left(\theta^{(1)}, \theta^{(2)}, \cdots, \theta^{(t)}\right) \in \Theta^t$, which are called *executions* of the algorithm. $\Theta^t$ is the $t$-fold Cartesian product of $\Theta$ with itself. Due to the randomness in the algorithm, this will result in a distribution $\mathcal{P}_{\mathcal{G}}$ over $\Theta^t$. Moreover, there is an adversary with access to the communication channels between the players. Therefore, for any execution $\boldsymbol{\theta} \in \Theta^t$, the adversary can observe $\mathcal{R}_{\mathcal{G}}(\boldsymbol{\theta}) = \left(o^{(1)}, o^{(2)}, \cdots, o^{(t)}\right) =: \boldsymbol{o} \in \mathcal{O}^t$, where $\mathcal{O}$ is the space of all *observations* at a single timestep. Hence, we can define the $(\epsilon, \delta)$-differential privacy ($(\epsilon, \delta)$-DP) as follows.

**Definition 2.2** ($(\epsilon, \delta)$-Differential Privacy). For an $\epsilon \geq 0$ and $\delta \geq 0$, an iterative distributed algorithm for finding equilibria is $(\epsilon, \delta)$-differentially private, if and only if for any two adjacent polymatrix game $\mathcal{G}, \mathcal{G}'$, any timestep $t > 0$ and any set of observations $\mathcal{S} \subseteq \mathcal{O}^t$,

$$\mathcal{P}_{\mathcal{G}}\left(\{\boldsymbol{\theta} : \mathcal{R}_{\mathcal{G}}(\boldsymbol{\theta}) \in \mathcal{S}\}\right) \leq e^{\epsilon} \mathcal{P}_{\mathcal{G}'}\left(\{\boldsymbol{\theta} : \mathcal{R}_{\mathcal{G}'}(\boldsymbol{\theta}) \in \mathcal{S}\}\right) + \delta. \tag{1}$$

Intuitively, Definition 2.2 guarantees that the algorithm will generate similar observations while deployed on adjacent games. Therefore, the adversary cannot differentiate $\mathcal{G}, \mathcal{G}'$ from the observation so he cannot infer the utility matrices with certainty from observations.

In this paper, we consider a generalization of Definition 2.2, Rényi differential privacy (Rényi DP) (Mironov, 2017). The distribution of observations up to timestep $t$ can be written as $\mu_{\mathcal{G}}^{(t)}(\boldsymbol{o}) := \mathcal{P}_{\mathcal{G}}\left(\mathcal{R}_{\mathcal{G}}^{-1}(\boldsymbol{o})\right)$ for $\boldsymbol{o} \in \mathcal{O}^t$ and thus we can define the $(\alpha, \epsilon)$-Rényi DP as follows.

**Definition 2.3** ($(\alpha, \epsilon)$-Rényi Differential Privacy). For $\alpha > 1$ and $\epsilon \geq 0$, an iterative distributed algorithm for finding equilibria is $(\alpha, \epsilon)$-Rényi differentially private, if and only if for any two adjacent polymatrix game $\mathcal{G}, \mathcal{G}'$ and timestep $t > 0$,

$$D_{\alpha}\left(\mu_{\mathcal{G}}^{(t)}, \mu_{\mathcal{G}'}^{(t)}\right) := \frac{1}{\alpha - 1} \log \mathbb{E}_{\boldsymbol{o} \sim \mu_{\mathcal{G}'}^{(t)}} \left[\left(\frac{\mu_{\mathcal{G}}^{(t)}(\boldsymbol{o})}{\mu_{\mathcal{G}'}^{(t)}(\boldsymbol{o})}\right)^{\alpha}\right] \leq \epsilon, \tag{2}$$

where $\epsilon$ is also called the *privacy budget*.

Definition 2.3 can be extended to $\alpha = 1$ and $\alpha = +\infty$ by defining $D_1(\cdot, \cdot) = \lim_{\alpha \to 1^+} D_{\alpha}(\cdot, \cdot)$, which is KL-divergence (Mironov, 2017), and $D_{\infty}(\cdot, \cdot) := \lim_{\alpha \to +\infty} D_{\alpha}(\cdot, \cdot)$. Moreover, Rényi DP can be converted to $(\epsilon, \delta)$-DP according to the following lemma from Mironov (2017).

**Lemma 2.4.** *If an algorithm satisfies $(\alpha, \epsilon)$-Rényi DP for $\alpha > 1$, then for any $\delta \in (0, 1)$, the algorithm also satisfies $\left(\epsilon + \frac{\log(1/\delta)}{\alpha - 1}, \delta\right)$-DP. Moreover, $(\infty, \epsilon)$-Rényi DP is equivalent to $(\epsilon, 0)$-DP.*

In light of Lemma 2.4, we focus on Rényi DP throughout the remainder of the paper.

## 3   Necessary Conditions for Achieving High Accuracy and Differential Privacy Simultaneously

In this section, we discuss the necessary conditions to achieve differential privacy and high accuracy in approximating equilibria in polymatrix games. Specifically, we will show that: (i) the adversary should not have access to all communication channels; and (ii) the accuracy metric cannot be the Euclidean distance to the equilibrium. If even one of the conditions is violated, one can find a constant

$c_0 > 0$ so that when $N$ is large enough, there exists a polymatrix game with $N$ players so that the approximation error $\zeta$ and differential privacy budget $\epsilon$ cannot be guaranteed to be smaller than $c_0$ simultaneously in that game. The results are summarized in Figure 1 (b).

All hardness results in this section are constructed with zero-sum polymatrix games (Cai et al., 2016), a special class of polymatrix games, that is, polymatrix games $\mathcal{G} = \left([N], E, \{\mathcal{A}_i\}_{i \in [N]}, \{\boldsymbol{U}_{i,j}\}_{(i,j) \in E}\right)$ such that, for any strategy profile $\boldsymbol{\pi}$, $\sum_{i \in [N]} \sum_{j \in \mathcal{N}(i)} \pi_i^\top \boldsymbol{U}_{i,j} \pi_j \equiv 0$. Since zero-sum polymatrix game is a subclass of general-sum polymatrix game, the hardness result also holds for general-sum polymatrix games automatically.

One can see that when $\boldsymbol{U}_{i,j} = -\boldsymbol{U}_{j,i}$ for any $(i,j) \in E$, the polymatrix game is zero-sum. Since the coarse correlated equilibria (CCE) in zero-sum polymatrix games collapse to NE (Cai et al., 2016), an algorithm that approximates CCE will approximate NE. Therefore, without loss of generality, in this section, we assume the algorithm will approximate the NE instead of CCE.

### 3.1 NECESSITY OF LIMITED ADVERSARIAL CHANNEL ACCESS

Assume that there is a distributed algorithm for finding equilibrium with the following guarantees of accuracy and privacy. In any polymatrix game $\mathcal{G}$ with $N$ players, the algorithm outputs a strategy profile $\boldsymbol{\pi}$ with distribution $\mu_{\mathcal{G}}$ on the observations. Here we omit the timestep superscript on $\mu_{\mathcal{G}}$ to emphasize that it is the distribution over all observations till termination of the algorithm. Then, there exists a parameter $\alpha \geq 1$ such that the algorithm satisfies the following accuracy and privacy guarantees in any $N$-player game $\mathcal{G}$:

$$\text{Accuracy: } \frac{1}{N} \sum_{i=1}^{N} \mathbb{E}\left[\max_{\widehat{\pi}_i \in \Delta^{\mathcal{A}_i}} \langle \pi_i - \widehat{\pi}_i, \boldsymbol{g}_i^{\boldsymbol{\pi}} \rangle\right] \leq \zeta \tag{3}$$

$$\text{Privacy: } D_\alpha\left(\mu_{\mathcal{G}}, \mu_{\mathcal{G}'}\right) \leq \epsilon \quad \forall \mathcal{G} \sim \mathcal{G}'. \tag{4}$$

When $\max_{\widehat{\pi}_i \in \Delta^{\mathcal{A}_i}} \langle \pi_i - \widehat{\pi}_i, \boldsymbol{g}_i^{\boldsymbol{\pi}} \rangle \leq \zeta$ for every player $i \in [N]$, the strategy profile $\boldsymbol{\pi}$ is called an $\zeta$-approximate NE. The accuracy metric $\max_{\widehat{\pi}_i \in \Delta^{\mathcal{A}_i}} \langle \pi_i - \widehat{\pi}_i, \boldsymbol{g}_i^{\boldsymbol{\pi}} \rangle$ is also known as *exploitability*. It measures how much player $i$ can benefit herself by unilaterally deviating from the current strategy profile $\boldsymbol{\pi}$, which should be 0 when $\boldsymbol{\pi}$ is the equilibrium. Exploitability provides a weaker notion of accuracy than Euclidean distance to the equilibrium: if $\boldsymbol{\pi}$ is close to an equilibrium in Euclidean distance, then its exploitability must be small, but the reverse need not hold.

In the following, we will show that if the adversary has access to all communication channels, it is impossible to achieve low exploitability $\zeta$ and low privacy budget $\epsilon$ simultaneously.

**Lemma 3.1.** *For any $N \geq 12$, there exists two zero-sum adjacent polymatrix games with $N$ players so that for any algorithm guaranteeing* (3) *and* (4), *we have*

$$\zeta \geq \min\left\{\frac{3\exp(-2\epsilon)}{112}, \frac{1}{112}\right\}. \tag{5}$$

Lemma 3.1 implies that even when $N$ goes to infinity, once $\zeta \leq \frac{1}{112}$, we have $\epsilon \geq \frac{\log 3}{2}$. To illustrate the proof of Lemma 3.1, we can first consider a relaxation of (3). If $\mathbb{E}\left[\max_{\widehat{\pi}_i \in \Delta^{\mathcal{A}_i}} \langle \pi_i - \widehat{\pi}_i, \boldsymbol{g}_i^{\boldsymbol{\pi}} \rangle\right] \leq \zeta$ holds for any player $i \in [N]$, then we can construct two adjacent polymatrix games, such that the approximate equilibrium any player $i \in [N]$ converges to differs much in two games, once the accuracy is lower than $\zeta$ *in both games*. Then, the adversary can distinguish those two games by computing the approximate equilibrium from his observations. While back to the original condition (3), we can use the pigeon-hole principle to construct a set of zero-sum polymatrix games, such that there exists two adjacent polymatrix games satisfying $\mathbb{E}\left[\max_{\widehat{\pi}_i \in \Delta^{\mathcal{A}_i}} \langle \pi_i - \widehat{\pi}_i, \boldsymbol{g}_i^{\boldsymbol{\pi}} \rangle\right] \leq \mathcal{O}(\zeta)$ simultaneously for some player $i \in [N]$. The full proof is postponed to Appendix B.1, where we also provide a lower bound for $(\epsilon, \delta)$-DP.

### 3.2 NECESSITY OF ACCURACY METRICS BEYOND EUCLIDEAN DISTANCE

Moreover, even if the adversary only has access to the observation of a single player, it is still impossible to find an accurate approximation of the equilibrium, in terms of Euclidean distance to the equilibrium, while guaranteeing privacy.

Let $\Delta^{\mathcal{A}_i,*}$ be the set of Nash equilibrium (NE) of player $i$ in a zero-sum polymatrix game. Moreover, $\mu_{\mathcal{G},i}$ denotes marginal distribution on the observation of player $i$ given the joint distribution $\mu_{\mathcal{G}}$ on all observations. Then, consider an algorithm with the following guarantee in any $N$-player game $\mathcal{G}$. There exists a parameter $\alpha \geq 1$ such that the algorithm satisfies that

$$\text{Accuracy:} \quad \frac{1}{N} \sum_{i=1}^{N} \mathbb{E}\left[ \|\pi_i - \text{Proj}_{\Delta^{\mathcal{A}_i,*}}(\pi_i)\|^2 \right] \leq \zeta \tag{6}$$

$$\text{Privacy:} \quad \frac{1}{N} \sum_{i=1}^{N} D_\alpha\left( \mu_{\mathcal{G},i}, \mu_{\mathcal{G}',i} \right) \leq \epsilon \quad \forall \mathcal{G} \sim \mathcal{G}'. \tag{7}$$

(6) guarantees that the output strategy of the algorithm is close to the set of NE, in terms of Euclidean distance. (7) states that for any two adjacent games $\mathcal{G} \sim \mathcal{G}'$, the Rényi divergence between the distribution of player $i$'s observations is small. This implies that by accessing the communication channel of player $i$, the adversary cannot distinguish $\mathcal{G}$ and $\mathcal{G}'$, so he cannot speculate the utility matrices.

**Lemma 3.2.** *For any $N \geq 8$, there exists two zero-sum adjacent polymatrix games with $N$ players so that for any algorithm guaranteeing (6) and (7), then*

$$\zeta \geq \min\left\{ \frac{3}{8} \exp\left(-4\epsilon\right), \frac{1}{16} \right\}. \tag{8}$$

Lemma 3.2 implies that once $\zeta \leq \frac{1}{16}$, the average differential privacy budget $\epsilon \geq \frac{\log 6}{4}$, even when $N$ goes to infinity. The proof of Lemma 3.2 is based on the following observation: for a two-player zero-sum game with row player's utility matrix as $\begin{bmatrix} 1 & 0 \\ 0 & 1 \end{bmatrix}$, when row player's strategy changes from $(0.51, 0.49)$ to $(0.49, 0.51)$, the column player's best-response changes from $(0, 1)$ to $(1, 0)$. In this way, for two adjacent polymatrix games with the utility matrices on the edge $(i, j)$ changes, player $i, j$'s equilibria will also change. Then, by a careful construction of the utility matrices in the game, such changes in equilibria will gradually propagate to the rest of the players. Lastly, we can prove that in two adjacent games with the equilibrium differs much, it is impossible to compute the equilibria accurately while the communications between players (the adversary's observations) are similar. The full proof can be found in Appendix B.2.

## 4 METHODS

In this section, we will propose an iterative algorithm for finding coarse correlated equilibrium (CCE) in general-sum polymatrix games (Moulin & Vial, 1978). As we will show in Section 6, this algorithm guarantees differential privacy under the conditions required in Section 3. We start by recalling a well-known connection between regret and convergence to CCE.

**Proposition 4.1.** *For an iterative algorithm that produces strategy profile $\boldsymbol{\pi}^{(t)}$ at timestep $t \in [T]$, its average strategy is the $\zeta$-approximate CCE of a polymatrix game $\mathcal{G}$ with $N$ players if and only if for any player $i \in [N]$, $\max_{\widehat{\pi}_i \in \Delta^{\mathcal{A}_i}} \frac{1}{T} \sum_{t=1}^{T} \left\langle \boldsymbol{g}_i^{\boldsymbol{\pi}^{(t)}}, \pi_i^{(t)} - \widehat{\pi}_i \right\rangle \leq \zeta$.*

By picking a timestep $t \in [T]$ uniformly at random and all players will play according to $\boldsymbol{\pi}^{(t)}$, the expected increase of utility when player $i$ deviates to $\widehat{\pi}_i$ is $\frac{1}{T} \sum_{t=1}^{T} \left\langle \boldsymbol{g}_i^{\boldsymbol{\pi}^{(t)}}, \pi_i^{(t)} - \widehat{\pi}_i \right\rangle$.

Let $\bar{\boldsymbol{g}}_i^{(t)} := \boldsymbol{g}_i^{\bar{\boldsymbol{\pi}}^{(t)}}$, $\boldsymbol{I}^{\mathcal{A}_i}$ as the identity matrix with index $\mathcal{A}_i \times \mathcal{A}_i$, and $\mathcal{N}\left(\boldsymbol{0}, \sigma^2 \boldsymbol{I}^{\mathcal{A}_i}\right)$ as the isotropic normal distribution over $\mathcal{A}_i$ with zero mean and $\sigma^2$ variance. Then, the algorithm is as follows: for any player $i \in [N]$, at each iteration $t \in \{0, 1, 2, \cdots, T-1, T\}$ for some $T > 0$ ($\pi_i^{(0)}$ is initialized as the uniform distribution over $\mathcal{A}_i$), player $i$ will sample a random noise $\boldsymbol{n}_i^{(t)} \sim \mathcal{N}\left(\boldsymbol{0}, \sigma^2 \boldsymbol{I}^{\mathcal{A}_i}\right)$. Then, she will broadcast her strategy $\pi_i^{(t)}$ plus the noise to her neighbors. After receiving the noisy strategies from her neighbors, she will do one step of the projected gradient descent with the gradient $\bar{\boldsymbol{g}}_i^{(t)}$, learning rate $\eta$, and weight-decay parameter $\tau_i$. The complete algorithm is shown in Algorithm 1.

---

**Algorithm 1** Differentially Private CCE Computation in Polymatrix Games

---

Input: Player index $i$

Initialize $\pi_i^{(0)}$ as uniform distribution over $\mathcal{A}_i$

Let $\overline{\mathcal{N}} := N \cdot (\sum_{i=1}^{N} \frac{1}{|\mathcal{N}(i)|})^{-1}$ be the harmonic mean of players' degrees

Initialize $\tau_i \leftarrow \frac{(\overline{\mathcal{N}})^{5/9}}{|\mathcal{N}(i)| \log N}$

**for** $t = 0, 1, ..., T$ **do**

    Sample $\boldsymbol{n}_i^{(t)} \sim \mathcal{N}\left(\boldsymbol{0}, \sigma^2 \boldsymbol{I}^{\mathcal{A}_i}\right)$

    Broadcast $\pi_i^{(t)} + \boldsymbol{n}_i^{(t)}$ to neighbors $j \in \mathcal{N}(i)$

    **for** $j \in \mathcal{N}(i)$ **do**

        Receive $\pi_j^{(t)} + \boldsymbol{n}_j^{(t)}$ and compute $\bar{\pi}_j^{(t)} \leftarrow \text{Proj}_{\Delta^{\mathcal{A}_j}}\left(\pi_j^{(t)} + \boldsymbol{n}_j^{(t)}\right)$

    **end for**

    Gradient $\bar{\boldsymbol{g}}_i^{(t)} \leftarrow -\frac{1}{|\mathcal{N}(i)|} \sum_{j \in \mathcal{N}(i)} \boldsymbol{U}_{i,j} \bar{\pi}_j^{(t)}$

$$\pi_i^{(t+1)} \leftarrow \underset{\pi_i \in \Delta^{\mathcal{A}_i}}{\arg\min} \left\langle \pi_i, \bar{\boldsymbol{g}}_i^{(t)} \right\rangle + \tau_i \|\pi_i\|^2 + \frac{1}{\eta} \left\|\pi_i - \bar{\pi}_i^{(t)}\right\|^2. \tag{9}$$

    {We remark that (9) is equivalent to $\pi_i^{(t+1)} \leftarrow \text{Proj}_{\Delta^{\mathcal{A}_i}}\left(\frac{\bar{\pi}_i^{(t)} - \eta \bar{\boldsymbol{g}}_i^{(t)}}{1 + \eta \tau_i}\right)$.}

**end for**

---

The intuition of $\tau_i \geq 0$ being inversely proportional to $|\mathcal{N}(i)|$ is that for players with fewer neighbors, her gradient is more sensitive to the variation of a single utility matrix, so more additional regularization (larger decay rate) is required to hide the difference of $\pi_i^{(t)}$ between adjacent games.

Moreover, to avoid observations from two adjacent games from increasingly diverging as updating more timesteps, the gradient descent of $\pi_i^{(t+1)}$ starts from $\bar{\pi}_i^{(t)}$ instead of $\pi_i^{(t)}$. However, we need to pay the price of suffering additional approximation error caused by the noise added to $\bar{\pi}_i^{(t)}$.

*Remark* 4.2. Let $\boldsymbol{n}^{(t)} := \left(\boldsymbol{n}_1^{(t)}, \boldsymbol{n}_2^{(t)}, \cdots, \boldsymbol{n}_N^{(t)}\right)$. In Algorithm 1, the execution of the algorithm at timestep $t$ is $\left(\boldsymbol{\pi}^{(t)}, \boldsymbol{n}^{(t)}\right)$. Then, $\mathcal{R}_{\mathcal{G}}\left(\left(\boldsymbol{\pi}^{(s)}, \boldsymbol{n}^{(s)}\right)_{s=1}^{t}\right) = \left(\bigcup_{i \in [N]} \left\{\pi_i^{(s)} + \boldsymbol{n}_i^{(s)}\right\}\right)_{s=1}^{t}$.

## 5 Convergence Analysis

This section will show the convergence of Algorithm 1 in general-sum polymatrix games.

**Theorem 5.1.** *Consider Algorithm 1. Let $A = \max_{i \in [N]} |\mathcal{A}_i|$. The update-rule can achieve the following guarantee in any polymatrix game. For any $T > 0$, player $i \in [N]$ and strategy $\pi_i \in \Delta^{\mathcal{A}_i}$,*

$$\frac{1}{NT} \sum_{i=1}^{N} \sum_{t=1}^{T} \mathbb{E}\left[\left\langle \boldsymbol{g}_i^{\boldsymbol{\pi}^{(t+1)}}, \pi_i^{(t+1)} - \pi_i \right\rangle\right]$$

$$\leq \frac{1}{\eta T} + A \frac{\sigma^2}{2\eta} + \left(\frac{2\eta^2}{\sigma} + \frac{7\sigma}{2}\right) A^{\frac{3}{2}} + \frac{1}{2\left(\overline{\mathcal{N}}\right)^{4/9} \log N} + \frac{2\eta\sqrt{A}}{\sigma \left(\overline{\mathcal{N}}\right)^{4/9} \log N}. \tag{10}$$

The proof sketch and the proof are postponed to Appendix C. Theorem 5.1 implies that the average strategy converges to coarse correlated equilibrium (CCE). Besides, it is noteworthy that when applying more noise for better differential privacy guarantees, *i.e.* larger $\sigma$, the convergence rate will worsen since the gradients provide less information.

With a fixed learning rate $\eta$ and noise $\sigma$, the accuracy increases when the number of players $N$ grows, while previous work (Gade et al., 2020; Ye et al., 2021; Wang et al., 2022; Shakarami et al., 2022) cannot, enabling achieving high accuracy and low differential budget simultaneously (see Section 7).

## 6 ANALYSIS OF DIFFERENTIAL PRIVACY

In this section, we show that Algorithm 1 can simultaneously guarantee differential privacy. Since Lemma 3.1 states that the adversary should not have access to all communication channels, we will bound Rényi DP when the adversary only has access to the communication channel of a single player, *i.e.* Rényi divergence of $\mu_{\mathcal{G},i}^{(t)}$, the marginal distribution of $\mu_{\mathcal{G}}^{(t)}$ over $\left(\pi_i^{(s)} + \boldsymbol{n}_i^{(s)}\right)_{s=1}^t$. According to the composition rule of Rényi differential privacy (Mironov, 2017), when the adversary has access to multiple communication channels, the privacy budget is the product of the number of communication channels he has access to, and the privacy budget when he only has access to a single communication channel. Effectively, this models a scenario where the communication network is largely reliable.

We break the analysis into two cases: dense graphs and sparse graphs. In the following, we will show that the privacy budget of Algorithm 1 is proportional to $\eta^2, T$ but inversely proportional to $\sigma^2$. Intuitively, when the learning rate $\eta$ is larger or the algorithm runs more timesteps, the observation will diverge more due to the different utility matrices. But when we add more noise, *i.e.* larger $\sigma$, it will be harder to determine which distribution a sample $\pi_i^{(t)} + \boldsymbol{n}_i^{(t)}$ is from.

**Theorem 6.1.** *Consider Algorithm 1 and any two adjacent polymatrix games $\mathcal{G}, \mathcal{G}'$ that differ at $(v_1, v_2)$. Let $dist(i, j)$ be the length of the shortest path between players $i, j$, which is $\infty$ when $i, j$ are not connected. The update-rule guarantees the following for any $T > 0$ and player $i \in [N]$,*

$$\frac{1}{N} \sum_{i=1}^N D_\alpha \left( \mu_{\mathcal{G},i}^{(T)}, \mu_{\mathcal{G}',i}^{(T)} \right) \le \frac{\alpha \eta^2}{\sigma^2} \min\{\clubsuit, \spadesuit\} T, \tag{11}$$

*where*

$$\clubsuit := \frac{16 A^3 (\log N)^2}{\left(\overline{\mathcal{N}}\right)^{4/9}} + \frac{4A}{N} \tag{12}$$

$$\spadesuit := \frac{2A}{N} \sum_{i=1}^N \mathbb{1}\left( T > \min\{dist(i, v_1), dist(i, v_2)\} \right). \tag{13}$$

The proof sketch and a complete proof can be found in Appendix D. Theorem 6.1 implies that when the graph is dense, *i.e.* $\overline{\mathcal{N}} \ge N^p$ for some $p \in (0, 1)$, $\clubsuit$ will be sublinear in $N$. When the graph is sparse, *i.e.* $\mathcal{N}^{\max} := \max_{i \in [N]} |\mathcal{N}(i)|$ is a constant, $\spadesuit$ will be small since the most players will be distant from $v_1, v_2$ (approximately $\Omega(\log N)$). Therefore, even with a fixed learning rate $\eta$ and noise $\sigma$, the privacy budget will decrease as the number of players grows, which is in stark contrast to previous work (Gade et al., 2020; Ye et al., 2021; Wang et al., 2022; Shakarami et al., 2022). The details are deferred to Section 7.

## 7 TRADE-OFF BETWEEN ACCURACY AND DIFFERENTIAL PRIVACY

According to Theorem 5.1 and Theorem 6.1, by picking $\eta, \sigma$ and $T$ carefully, when the number of players grows to infinity, we may obtain a zero privacy budget while converging to CCE.

**Theorem 7.1.** *Consider Algorithm 1 with $\sigma = \frac{1}{\sqrt{T}}$ and any two adjacent polymatrix games $\mathcal{G}, \mathcal{G}'$ that differs at $(v_1, v_2)$. The update-rule guarantees the following,*

$$\frac{1}{N} \sum_{i=1}^N \max \left\{ \frac{1}{T} \sum_{t=1}^T \mathbb{E}\left[ \left\langle \boldsymbol{g}_i^{\boldsymbol{\pi}^{(t+1)}}, \pi_i^{(t+1)} - \pi_i \right\rangle \right], 0 \right\}$$
$$\le \frac{A+1}{\eta T} + 2A^{3/2} \eta^2 \sqrt{T} + \frac{7A^{3/2}}{2\sqrt{T}} + \frac{1}{2 \left(\overline{\mathcal{N}}\right)^{4/9} \log N} + \frac{2\eta \sqrt{AT}}{\left(\overline{\mathcal{N}}\right)^{4/9} \log N} \tag{14}$$

$$\frac{1}{N} \sum_{i=1}^N D_\alpha \left( \mu_{\mathcal{G},i}^{(T)}, \mu_{\mathcal{G}',i}^{(T)} \right) \le \alpha \eta^2 \min\{\clubsuit, \spadesuit\}. \tag{15}$$

The proof is simply substituting the values of $\sigma$ into Theorem 5.1 and Theorem 6.1. When the graph is dense, *i.e.* $\overline{\mathcal{N}} \geq N^p$ for some $p > 0$, take $\eta = \frac{1}{T\clubsuit^{1/3}}, T = \frac{N^{8p/9}}{(\log N)^4}$, then

$$\frac{1}{NT} \sum_{i=1}^N \sum_{t=1}^T \mathbb{E}\left[\left\langle g_i^{\pi^{(t+1)}}, \pi_i^{(t+1)} - \pi_i \right\rangle\right] \leq \mathcal{O}\left(\frac{(\log N)^{2/3}}{N^{4p/27}}\right),$$

where the $\mathcal{O}$ notation takes $A$ as constant. Simultaneously, the differential budget guarantees that

$$\frac{1}{N} \sum_{i=1}^N D_\alpha\left(\mu_{\mathcal{G},i}^{(T)}, \mu_{\mathcal{G}',i}^{(T)}\right) \leq \mathcal{O}\left(\frac{\alpha (\log N)^{2/3}}{N^{4p/27}}\right).$$

Therefore, when the graph is dense, the accuracy and differential privacy budget of Algorithm 1 would both be better as the number of players increases.

Consider when the graph is sparse, *i.e.* $\mathcal{N}^{\max}$ is a constant. If $\mathcal{N}^{\max} \geq 2$, let $\eta = \frac{1}{T\spadesuit^{1/3}}, T = (1 - \log_N \log N) \log_{\mathcal{N}^{\max}} N$, then $\spadesuit \leq \frac{4A}{\log N}$, since at most $\frac{2(\mathcal{N}^{\max})^T}{\mathcal{N}^{\max}-1} \leq \frac{2N}{\log N}$ players satisfy $\min\{\text{dist}(i, v_1), \text{dist}(i, v_2)\} < T$. If $\mathcal{N}^{\max} = 1$, $\spadesuit \leq \frac{2}{N}$ for any $T > 0$. Therefore,

$$\frac{1}{NT} \sum_{i=1}^N \sum_{t=1}^T \mathbb{E}\left[\left\langle g_i^{\pi^{(t+1)}}, \pi_i^{(t+1)} - \pi_i \right\rangle\right] \leq \mathcal{O}\left(\frac{1}{(\log N)^{1/3}}\right)$$

$$\frac{1}{N} \sum_{i=1}^N D_\alpha\left(\mu_{\mathcal{G},i}^{(T)}, \mu_{\mathcal{G}',i}^{(T)}\right) \leq \mathcal{O}\left(\frac{\alpha}{(\log N)^{1/3}}\right).$$

Compared to previous work (Gade et al., 2020; Ye et al., 2021; Wang et al., 2022; Shakarami et al., 2022), the main contribution of Theorem 7.1 is that for any desired accuracy $\zeta_0 > 0$ and differential privacy budget $\epsilon_0 > 0$, we can ensure our accuracy and differential privacy budget to be lower than $\zeta_0, \epsilon_0$ simultaneously when $N$ is large enough. In contrast, to keep accuracy smaller than $\zeta_0$, previous work will suffer a differential privacy budget larger than $\epsilon_0$, no matter how large $N$ is, and vice versa.

We emphasize that Theorem 7.1 bounds the differential privacy budget when the adversary can access a single communication channel. By the composition rule for Rényi differential privacy (Mironov, 2017), see the beginning of Section 6, the overall privacy budget scales linearly with the number of channels the adversary accesses, specifically, it equals the number of accessed channels times the bound in Theorem 7.1.

In the non-private setting, the noise $\sigma$ and the additional regularization $\tau$ can be removed. Consequently, the algorithm reduces to Online Mirror Descent with a Euclidean norm regularizer (Hazan et al., 2016), achieving a convergence rate of $\mathcal{O}\left(\frac{1}{\eta T} + \eta\right)$. However, in the private setting, the terms $\mathcal{O}\left(\frac{(\log N)^{2/3}}{N^{4p/27}}\right)$ and $\mathcal{O}\left(\frac{1}{(\log N)^{1/3}}\right)$ represent the error incurred within $T$ iterations, rather than the convergence rate itself. Given that $T = \frac{N^{8p/9}}{(\log N)^4}$ and $T = (1 - \log_N \log N) \log_{\mathcal{N}^{\max}} N$ respectively, the algorithm maintains a rapid convergence rate.

## 8 EXPERIMENTS

In this section, we evaluate the exploitability $\left(\max_{\widehat{\pi}_i \in \Delta^{\mathcal{A}_i}} \langle \pi_i - \widehat{\pi}_i, g_i^\pi \rangle\right)$ of our method and a baseline algorithm obtained by adapting Huang et al. (2015) from an single-objective optimization formulation to the game setting. As shown in Figure 3 and Figure 4, under the same differential-privacy budget, our method consistently attains lower exploitability. Moreover, as the number of players increases, the exploitability of our method decreases, whereas the baseline's does not.

Figure 3 shows the results for an Erdős–Rényi graph, where each pair of players interacts with probability $p$.

Figure 4 displays the results for graphs with a clustered topology. Specifically, players are randomly divided into $\lfloor 1/p \rfloor$ clusters. All pairs of players within the same cluster interact with one another, while pairs of players belonging to different clusters interact with probability $\frac{10p}{N}$. This setup models scenarios characterized by dense within-cluster interactions. Additional experimental results are provided in Appendix E.

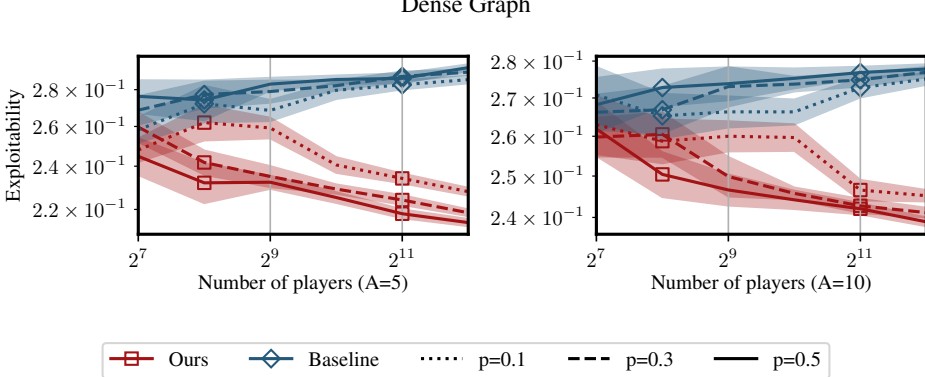

Figure 3: We evaluate the exploitability of our algorithm and a baseline on dense graphs. Each node (player) connects to another node independently with probability $p$, and duplicate edges are then removed. All players have action sets of size $A$. Both the baseline and our method are run under the same differential privacy budget.

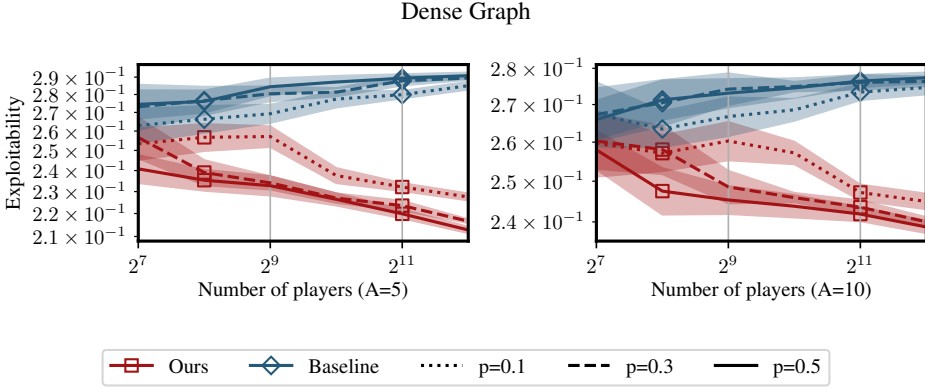

Figure 4: The exploitability of our algorithm and the baseline for dense graphs with cluster topology. All players have action sets of size $A$. Both the baseline and our method are run under the same differential privacy budget.

## 9 CONCLUSION AND LIMITATIONS

This paper shows necessary conditions to find the equilibria of polymatrix games with DP guarantees. Moreover, we propose an algorithm to find the equilibria with DP guarantees under the conditions. Moreover, as the number of players in the game increases, the algorithm will gradually achieve perfect accuracy and DP guarantees. Lastly, we provide empirical evidence to justify the algorithm in Appendix E. Currently, our framework is specified for polymatrix games. Therefore, an interesting future direction is to extend it to other games, such as normal-form games and extensive-form games.

## ACKNOWLEDGEMENT

The authors thank the anonymous reviewers and the area chair for their helpful feedback. M.L. was supported by the MathWorks Fellowship. A.O. and G.F. were supported in part by the ONR grant N000142512296. G.F. was additionally supported by CCF-2443068 and an AI2050 Early Career Fellowship.

## 10 ETHICS STATEMENT

This paper presents work that aims to advance the field of Machine Learning. There are many potential societal consequences of our work, none of which we feel must be specifically highlighted here.

## 11 REPRODUCIBILITY STATEMENT

The proofs are presented in Appendices B to D.

## 12 USE OF LARGE LANGUAGE MODELS

In this paper, we use large language models (LLMs). For example, by correcting grammatical errors, generating illustrative images (Figure 1), and producing code.

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

# A   RELATED WORK

In this section, we discuss prior work specifically targeted at distributed *equilibrium computation*. We omit a discussion of prior work on distributed (single-agent) optimization as it is hardly applicable to our purposes, for several important reasons. For one, distributed optimization with differential privacy guarantee (Huang et al., 2015; Nozari et al., 2016; Wei et al., 2020; Cyffers et al., 2022; Mrini et al., 2024) aims to find the minimum of the weighted sum of players' utility functions. Specifically, the adversary cannot distinguish between two sets of functions that differ only in one player's function based on his observations, as discussed in (Cyffers et al., 2022; Mrini et al., 2024). However, distributed equilibrium finding is more complex since it aims to find the equilibrium point of those utility functions. Furthermore, the target output of the distributed optimization is of constant dimensionality with respect to the number of agents involved in the optimization process, such as the best parameters for a neural network (Wei et al., 2020). However, the output's dimension of equilibrium finding scales linearly with respect to the number of agents, since we need to specify the strategy of each individual player.

**Online Learning with Differential Privacy.**   Prior work, including Jain et al. (2012); Guha Thakurta & Smith (2013); Agarwal & Singh (2017), has studied no-regret learning with DP guarantee, focusing on ensuring that an adversary cannot determine the utility (loss) vector at any single timestep. However, that condition does not imply differential privacy with respect to the payoff function itself. For instance, in a two-player normal-form game, the utility vector at timestep $t$ for player 1 is $\boldsymbol{U}\pi_2^{(t)}$, where $\boldsymbol{U}$ is the utility matrix and $\pi_2^{(t)}$ is the strategy of player 2 at timestep $t$. In this case, an algorithm that reveals $\boldsymbol{U}\pi_2^{(t)} + \boldsymbol{n}^{(t)}$, where $\boldsymbol{n}^{(t)}$ is sampled from a multivariate Gaussian, ensures differential privacy with respect to each individual utility vector. However, consider the case when $\pi_2^{(t)}$ is sampled uniformly on each axis for any timestep $t$. By taking the average over all utility vectors, the Gaussian noise cancels out, and we will get the average of $\boldsymbol{U}$'s columns. Therefore, $\boldsymbol{U}$ is not differentially private, while each utility vector is still private.

**Differential Privacy in Minimax Optimization.**   Yang et al. (2022); Zhang et al. (2022); Bassily et al. (2023); Boob & Guzmán (2024) analyze differential privacy in minimax optimization, but their setting differs from ours in several aspects. First, they study two-player zero-sum games, whereas we consider multi-player general-sum polymatrix games (a class that subsumes two-player zero-sum). Second, they assume each player's utility decomposes as an average of multiple component utilities and aim to prevent an adversary from inferring any single component. This, however, does not protect the average itself, the true game utility, which is precisely what we seek to keep private.

**Differential Privacy in Games.**   Other prior work, including Gade et al. (2020); Ye et al. (2021); Wang et al. (2022); Shakarami et al. (2022), has focused on finding equilibrium in aggregative games with differential privacy guarantees. However, they cannot achieve accuracy and differential privacy simultaneously (*e.g.,* privacy budget $\mathcal{O}\left(\epsilon\right)$ and accuracy $\mathcal{O}\left(\frac{1}{\epsilon}\right)$). In other words, to achieve a fixed accuracy, the privacy budget will not continue to decrease while the number of players increases. The reason is that they did not exploit the game's structure to cancel out the noise imposed during the update. Therefore, accuracy will be harmed when achieving a low differential privacy budget by adding a large amount of noise. Furthermore, the privacy guarantee of Wang & Nedić (2024); Wang & Başar (2024) is weak because the adversary may determine the value of the utility function in a small region around the equilibrium point. Therefore, when the utility function is multilinear, such as polymatrix games, normal-form games, and extensive-form games, the adversary can fully determine the utility functions with certainty. Instead, in this paper, we exploit the structure of polymatrix games, whether the associated graph is dense or sparse, to mitigate the impact of the imposed noise, thereby achieving accuracy and differential privacy simultaneously. Specifically, we can achieve privacy budget $\mathcal{O}\left(\frac{\epsilon}{f(N)}\right)$ and accuracy $\mathcal{O}\left(\frac{1}{\epsilon g(N)}\right)$, where $f(N), g(N)$ are monotone functions with respect to the number of players $N$ and goes to infinity as $N \to \infty$. Details can be found in Theorem 7.1.

## B    PROOF OF SECTION 3

The section is arranged as follows. For both Lemma 3.1 and Lemma 3.2, we will show that there exists a polymatrix game, so that achieving accuracy and $(\epsilon, \delta)$-DP simultaneously is impossible. Then, we further show that achieving accuracy and Rényi DP is impossible by showing that for $(1, \epsilon)$-Rényi DP. Because the $\alpha$-Rényi divergence grows monotonically with respect to $\alpha$ (Mironov, 2017).

### B.1    PROOF OF LEMMA 3.1

**Lemma 3.1.** *For any $N \geq 12$, there exists two zero-sum adjacent polymatrix games with $N$ players so that for any algorithm guaranteeing* (3) *and* (4)*, we have*

$$\zeta \geq \min\left\{ \frac{3\exp(-2\epsilon)}{112}, \frac{1}{112} \right\}. \tag{5}$$

Table 1: $i$ is the row player and $j$ is the column player, the utility in the table is $\boldsymbol{U}_{i,j}(a_i, a_j) = -\boldsymbol{U}_{j,i}(a_j, a_i)$.

|       | $a_1$ |       |       | $a_1$ |       |       | $a_1$ | $a_2$ |
|-------|-------|-------|-------|-------|-------|-------|-------|-------|
| $a_1$ | 1     |       | $a_1$ | 0     |       | $a_1$ | 0.5   | 0     |
| $a_2$ | 0     |       | $a_2$ | 1     |       | $a_2$ | 0     | 0.5   |

*Proof.* Consider a *zero-sum* polymatrix game with $3N$ players, where its corresponding graph is a chain, *i.e.* the edge set $E = \{(i, i+1)\}_{i\in[3N-1]}$. For $i \in [N]$, $\boldsymbol{U}_{3i-1,3i-2}$ is Table 1 (left) and $\boldsymbol{U}_{3i,3i-1}$ is Table 1 (right). All other utility matrices are zero matrices. Therefore, the only equilibrium is all players playing $a_1$. Let denote the game as $\mathcal{G}^\emptyset$.

In this case, for any $i \in [N]$, when changing $\boldsymbol{U}_{3i-1,3i-2}$ to Table 1 (middle), the equilibrium of both $3i - 1$ and $3i$ will be choosing $a_2$ deterministically, and the equilibrium of other players keep unchanged. Let denote this new game as $\mathcal{G}^{\{i\}}$.

#### B.1.1    LOWERBOUNDS ON $(\epsilon, \delta)$-DP

This section will show that if the algorithm satisfies $(\epsilon, \delta)$-DP, then

$$\zeta \geq \frac{1-\delta}{28\left(1+e^\epsilon\right)}. \tag{16}$$

Suppose the distributed algorithm satisfies the following guarantee of accuracy and privacy. Let $\boldsymbol{g}_i^{\boldsymbol{\pi}}(\mathcal{G})$ be the gradient of player $i$ in $\mathcal{G}$ and $\boldsymbol{\pi}^{\{i\}}$ be the output of the algorithm in $\mathcal{G}^{\{i\}}$. Then,

$$\forall\, i \in [N],\ \sum_{j=1}^{3N} \mathbb{E}\left[ \max_{\widehat{\pi}_j \in \Delta^{\mathcal{A}_j}} \left\langle \pi_j^{\{i\}} - \widehat{\pi}_j, \boldsymbol{g}_j^{\boldsymbol{\pi}^{\{i\}}}(\mathcal{G}^{\{i\}}) \right\rangle \right] \leq N\zeta \qquad \text{(Accuracy)}$$

$$\forall\, i \in [N],\ \text{set } \mathcal{C},\quad \Pr(\boldsymbol{\pi}^\emptyset \in \mathcal{C}) \leq e^\epsilon \Pr(\boldsymbol{\pi}^{\{i\}} \in \mathcal{C}) + \delta, \qquad \text{(Privacy)}$$

Moreover, it is easy to verify that for any strategy profile $\boldsymbol{\pi} \in \bigtimes_{i=1}^{3N} \Delta^{\mathcal{A}_i}$ and $i \in [N], k \in \{0\} \cup [N]$ (let $\mathcal{G}^{\{0\}} = \mathcal{G}^\emptyset$ for simplicity), we have

$$\max_{\widehat{\pi}_{3i-1}\in\Delta^{\mathcal{A}_{3i-1}}} \left\langle \pi_{3i-1} - \widehat{\pi}_{3i-1}, \boldsymbol{g}_{3i-1}^{\boldsymbol{\pi}}(\mathcal{G}^{\{k\}}) \right\rangle + \max_{\widehat{\pi}_{3i}\in\Delta^{\mathcal{A}_{3i}}} \left\langle \pi_{3i} - \widehat{\pi}_{3i}, \boldsymbol{g}_{3i}^{\boldsymbol{\pi}}(\mathcal{G}^{\{k\}}) \right\rangle \geq \begin{cases} 0.5\pi_{3i}(a_2) & i \neq k \\ 0.5\pi_{3i}(a_1) & i = k. \end{cases}$$

Since the algorithm is distributed, the findr for each player can be written as $f_i\colon \mathcal{O}^T \times \mathbb{R}^{\mathcal{A}_i \times \mathcal{A}_{i-1}} \times \mathbb{R}^{\mathcal{A}_i \times \mathcal{A}_{i+1}} \times \Delta^{\mathcal{A}_i} \to \mathbb{R}$ (let $\mathcal{A}_0 = \mathcal{A}_{N+1} = \emptyset$ for notational simplicity), which is a conditional distribution over all possible strategies in $\Delta^{\mathcal{A}_i}$, conditioned on all past observations and player's utility matrices.

Let $\text{adv}(\boldsymbol{o}, \pi_{3i}) := f_{3i}(\boldsymbol{o}_{3i}, \boldsymbol{U}_{3i,3i-1}, \pi_{3i})$ for $\boldsymbol{o} \in \mathcal{O}^T$ and $\pi_{3i} \in \Delta^{\mathcal{A}_{3i}}$, where $\boldsymbol{U}_{3i,3i-1}$ is the utility matrix between player $3i, 3i-1$ in $\mathcal{G}^\emptyset$. Note that since the adversary cannot access utility matrices, $\text{adv}(\boldsymbol{o}, \pi_{3i})$ is fixed even though the observation now comes from games $\mathcal{G}^{\{k\}}$ for some $k \in [N]$.

Then, the adversary may sample $\pi_{3i}^{\text{adv,k}}$ according to $\text{adv}(\boldsymbol{o}^k, \pi_{3i})$, where $\boldsymbol{o}^k \in \mathcal{O}^T$ is the observation in game $\mathcal{G}^{\{k\}}$. Because the utility matrices of player $3i$ in $\mathcal{G}^\emptyset$ and $\mathcal{G}^{\{k\}}$ are identical, we have

$$0.5\mathbb{E}\left[\pi_{3i}^{\text{adv},k}(a_2)\right] = 0.5\mathbb{E}\left[\pi_{3i}^{\{k\}}(a_2)\right]$$
$$\leq \mathbb{E}\left[\max_{\widehat{\pi}_{3i-1}\in\Delta^{\mathcal{A}_{3i-1}}} \left\langle \pi_{3i-1}^{\{k\}} - \widehat{\pi}_{3i-1}, \boldsymbol{g}_{3i-1}^{\boldsymbol{\pi}^{\{k\}}}(\mathcal{G}^{\{k\}})\right\rangle + \max_{\widehat{\pi}_{3i}\in\Delta^{\mathcal{A}_{3i}}} \left\langle \pi_{3i}^{\{k\}} - \widehat{\pi}_{3i}, \boldsymbol{g}_{3i}^{\boldsymbol{\pi}^{\{k\}}}(\mathcal{G}^{\{k\}})\right\rangle\right].$$

Similar bounds also holds for $\mathbb{E}\left[\pi_{3i}^{\text{adv}}(a_1)\right]$ when $i = k$.

Then, by Markov inequality and the definition of $(\epsilon, \delta)$-DP, we have

$$4\mathbb{E}\left[\max_{\widehat{\pi}_{3i-1}\in\Delta^{\mathcal{A}_{3i-1}}} \left\langle \pi_{3i-1}^\emptyset - \widehat{\pi}_{3i-1}, \boldsymbol{g}_{3i-1}^{\boldsymbol{\pi}^\emptyset}(\mathcal{G}^\emptyset)\right\rangle + \max_{\widehat{\pi}_{3i}\in\Delta^{\mathcal{A}_{3i}}} \left\langle \pi_{3i}^\emptyset - \widehat{\pi}_{3i}, \boldsymbol{g}_{3i}^{\boldsymbol{\pi}^\emptyset}(\mathcal{G}^\emptyset)\right\rangle\right]$$
$$\geq 2\mathbb{E}\left[\pi_{3i}^{\text{adv},\emptyset}(a_2)\right]$$
$$\geq \Pr\left(\pi_{3i}^{\text{adv},\emptyset}(a_2) \geq 0.5\right)$$
$$= \Pr\left(\pi_{3i}^{\text{adv},\emptyset}(a_1) \leq 0.5\right) \tag{17}$$
$$\geq e^{-\epsilon}\Pr\left(\pi_{3i}^{\text{adv},\{i\}}(a_1) \leq 0.5\right) - \delta e^{-\epsilon}$$
$$\geq e^{-\epsilon}\left(1 - 2\mathbb{E}\left[\pi_{3i}^{\text{adv},\{i\}}(a_1)\right]\right) - \delta e^{-\epsilon}$$
$$\geq e^{-\epsilon}\left(1 - 4\mathbb{E}\left[\max_{\widehat{\pi}_{3i-1}\in\Delta^{\mathcal{A}_{3i-1}}} \left\langle \pi_{3i-1}^{\{i\}} - \widehat{\pi}_{3i-1}, \boldsymbol{g}_{3i-1}^{\boldsymbol{\pi}^{\{i\}}}(\mathcal{G}^{\{i\}})\right\rangle + \max_{\widehat{\pi}_{3i}\in\Delta^{\mathcal{A}_{3i}}} \left\langle \pi_{3i}^{\{i\}} - \widehat{\pi}_{3i}, \boldsymbol{g}_{3i}^{\boldsymbol{\pi}^{\{i\}}}(\mathcal{G}^{\{i\}})\right\rangle\right]\right) - \delta e^{-\epsilon}.$$

Moreover, for each $i \in [N]$ and $\mathcal{G}^{\{i\}}$, we can further define $\mathcal{G}^{\{i,j\}}$ for $i \neq j \in [N]$ as the adjacent game to $\mathcal{G}^{\{i\}}$ that differs from $\mathcal{G}^{\{i\}}$ only at $\boldsymbol{U}_{3j-1,3j-2}$, by changing it from Table 1 (left) to Table 1 (middle). Also note that $\mathcal{G}^{\{i,j\}}$ is also adjacent to $\mathcal{G}^{\{j\}}$. Such a process can be recursively applied to $\mathcal{G}^{\{i,j\}}$ and finally we will get $2^N$ games $\left\{\mathcal{G}^{\mathcal{S}}\right\}_{\mathcal{S}\subseteq[N]}$.

**Lemma B.1.** *Consider the $2^N$ games $\left\{\mathcal{G}^{\mathcal{S}}\right\}_{\mathcal{S}\subseteq[N]}$ constructed above. There must exist two adjacent games $\mathcal{G}^{\mathcal{S}}$ and $\mathcal{G}^{\mathcal{S}\cup\{i\}}$ with $i \in [N] \setminus \mathcal{S}$, so that*

$$\mathbb{E}\left[\max_{\widehat{\pi}_{3i-1}\in\Delta^{\mathcal{A}_{3i-1}}} \left\langle \pi_{3i-1}^{\mathcal{S}} - \widehat{\pi}_{3i-1}, \boldsymbol{g}_{3i-1}^{\boldsymbol{\pi}^{\mathcal{S}}}(\mathcal{G}^{\mathcal{S}})\right\rangle + \max_{\widehat{\pi}_{3i}\in\Delta^{\mathcal{A}_{3i}}} \left\langle \pi_{3i}^{\mathcal{S}} - \widehat{\pi}_{3i}, \boldsymbol{g}_{3i}^{\boldsymbol{\pi}^{\mathcal{S}}}(\mathcal{G}^{\mathcal{S}})\right\rangle\right] \leq 7\zeta \tag{18}$$

$$\mathbb{E}\left[\max_{\widehat{\pi}_{3i-1}\in\Delta^{\mathcal{A}_{3i-1}}} \left\langle \pi_{3i-1}^{\mathcal{S}\cup\{i\}} - \widehat{\pi}_{3i-1}, \boldsymbol{g}_{3i-1}^{\boldsymbol{\pi}^{\mathcal{S}\cup\{i\}}}(\mathcal{G}^{\mathcal{S}\cup\{i\}})\right\rangle + \max_{\widehat{\pi}_{3i}\in\Delta^{\mathcal{A}_{3i}}} \left\langle \pi_{3i}^{\mathcal{S}\cup\{i\}} - \widehat{\pi}_{3i}, \boldsymbol{g}_{3i}^{\boldsymbol{\pi}^{\mathcal{S}\cup\{i\}}}(\mathcal{G}^{\mathcal{S}\cup\{i\}})\right\rangle\right] \leq 7\zeta. \tag{19}$$

The proof is postponed to Appendix B.3. Since (17) also holds for the $\mathcal{G}^{\mathcal{S}}$ and $\mathcal{G}^{\mathcal{S}\cup\{i\}}$ in Lemma B.1, we have

$$28\left(e^{-\epsilon} + 1\right)\zeta \geq e^{-\epsilon} - \delta e^{-\epsilon}.$$

Therefore, $\zeta \geq \frac{e^{-\epsilon} - \delta e^{-\epsilon}}{28(e^{-\epsilon}+1)} = \frac{1-\delta}{28(1+e^\epsilon)}$.

### B.1.2 LOWERBOUNDS ON $(\alpha, \epsilon)$-RÉNYI DP

This section will show that if the algorithm satisfies $(\epsilon, \delta)$-DP, then

$$\frac{1}{N} \sum_{i=1}^{N} \zeta_i \geq \frac{1 - \max_{i \in [N]} \delta_i}{1 + \exp\left(\frac{1}{N} \sum_{i=1}^{N} \epsilon_i\right)} - \frac{2}{N}. \tag{20}$$

By monotonicity of Rényi Divergence (Mironov, 2017), we only need to show the lowerbound of KL-divergence. By further post-processing on $\pi_{3i}^{\mathrm{adv},\mathcal{S}}$ for $i \in [N]$ and $\mathcal{S} \subseteq [N]$, the adversary can output whether $\pi_{3i}^{\mathrm{adv},\mathcal{S}}(a_2) \geq 0.5$. Let the distribution of the output be $\mu_{3i}^{\mathcal{S}}$. Then, for the adjacent games $\mathcal{G}^{\emptyset}, \mathcal{G}^{\{i\}}$, where $i \in [N]$, we have

$$\epsilon \geq \mathrm{KL}\left(\mu_{3i}^{\emptyset}, \mu_{3i}^{\{i\}}\right) = \Pr\left(\pi_{3i}^{\mathrm{adv},\emptyset}(a_2) \geq 0.5\right) \log \frac{\Pr\left(\pi_{3i}^{\mathrm{adv},\emptyset}(a_2) \geq 0.5\right)}{\Pr\left(\pi_{3i}^{\mathrm{adv},\{i\}}(a_2) \geq 0.5\right)}$$

$$+ \left(1 - \Pr\left(\pi_{3i}^{\mathrm{adv},\emptyset}(a_2) \geq 0.5\right)\right) \log \frac{1 - \Pr\left(\pi_{3i}^{\mathrm{adv},\emptyset}(a_2) \geq 0.5\right)}{1 - \Pr\left(\pi_{3i}^{\mathrm{adv},\{i\}}(a_2) \geq 0.5\right)}$$

$$\overset{(i)}{\geq} (1 - \mathrm{expl}_i^{\emptyset}) \log \frac{1 - \mathrm{expl}_i^{\emptyset}}{\mathrm{expl}_i^{\{i\}}} + \mathrm{expl}_i^{\emptyset} \log \frac{\mathrm{expl}_i^{\emptyset}}{1 - \mathrm{expl}_i^{\{i\}}},$$

where

$$\mathrm{expl}_i^{\mathcal{S}} := 4\mathbb{E}\left[\max_{\widehat{\pi}_{3i-1} \in \Delta^{\mathcal{A}_{3i-1}}} \left\langle \pi_{3i-1}^{\mathcal{S}} - \widehat{\pi}_{3i-1}, \boldsymbol{g}_{3i-1}^{\boldsymbol{\pi}^{\mathcal{S}}}(\mathcal{G}^{\mathcal{S}}) \right\rangle + \max_{\widehat{\pi}_{3i} \in \Delta^{\mathcal{A}_{3i}}} \left\langle \pi_{3i}^{\mathcal{S}} - \widehat{\pi}_{3i}, \boldsymbol{g}_{3i}^{\boldsymbol{\pi}^{\mathcal{S}}}(\mathcal{G}^{\mathcal{S}}) \right\rangle\right].$$

$(i)$ is because $\Pr\left(\pi_{3i}^{\mathrm{adv},\emptyset}(a_2) \geq 0.5\right) \leq \mathrm{expl}_i^{\emptyset}$ and $\Pr\left(\pi_{3i}^{\mathrm{adv},\{i\}}(a_2) \geq 0.5\right) \geq 1 - \mathrm{expl}_i^{\{i\}}$.

By Lemma B.1, there exists $\mathcal{S} \subseteq [N]$ and $i \in [N] \setminus s$ so that $\mathrm{expl}_i^{\mathcal{S}}, \mathrm{expl}_i^{\mathcal{S} \cup \{i\}} \leq 28\zeta$. Therefore,

$$\epsilon \geq (1 - 28\zeta) \log \frac{1 - 28\zeta}{28\zeta} + 28\zeta \log \frac{28\zeta}{1 - 28\zeta} = (1 - 56\zeta) \log \frac{1 - 28\zeta}{28\zeta}.$$

It implies that when $\zeta \leq \frac{1}{112}$, $\epsilon \geq \frac{1}{2} \log \frac{3}{112\zeta}$ so that $\zeta \geq \frac{3}{112 \exp(2\epsilon)}$. $\qquad \square$

### B.2 PROOF OF LEMMA 3.2

**Lemma 3.2.** *For any $N \geq 8$, there exists two zero-sum adjacent polymatrix games with $N$ players so that for any algorithm guaranteeing (6) and (7), then*

$$\zeta \geq \min\left\{\frac{3}{8} \exp\left(-4\epsilon\right), \frac{1}{16}\right\}. \tag{8}$$

*Proof.* Consider a zero-sum polymatrix game, where its corresponding graph is a chain. Specifically, there is an edge $(i, j) \in E$, if and only if $i = j - 1$. Moreover, each player has two actions and $\boldsymbol{U}_{i,i+1}$ is shown in Table 2 (left).

Table 2: $i$ is the row player and $j$ is the column player, the utility in the table is $\boldsymbol{U}_i(a_i, a_j)$ and $\boldsymbol{U}_j(a_j, a_i) = -\boldsymbol{U}_i(a_i, a_j)$.

|  | $a_1$ | $a_2$ |  |  | $a_1$ | $a_2$ |
|---|---|---|---|---|---|---|
| $a_1$ | 0.5 | $0.5 - \iota_i$ |  | $a_1$ | $0.5 - 2\iota_i$ | $0.5 - 3\iota_i$ |
| $a_2$ | $0.5 - 3\iota_i$ | $0.5 - 2\iota_i$ |  | $a_2$ | $0.5 - \iota_i$ | 0.5 |

Moreover, $0.1 = \iota_1 = 10\iota_2 \geq 10^2 \iota_3 = \cdots = 10^{N-1} \iota_N > 0$. Then, the Nash equilibrium (NE) is $\left(a_1, a_2, a_1, a_2, \cdots, a_{\frac{(-1)^N + 3}{2}}\right)$. However, when the utility matrix between player $(1, 2)$ changes to

Table 2 (right), the NE becomes $\left(a_2, a_1, a_2, a_1, \cdots, a_{\frac{(-1)^{N+1}+3}{2}}\right)$. Let the original game be denoted as $\mathcal{G}^0$ and the new game as $\mathcal{G}^1$.

With the observation $\boldsymbol{o}_i \in \mathcal{O}^T$ of any player $i > 2$, the adversary may sample strategy $\pi_i^{\mathrm{adv}}$ from the distribution $\mathrm{adv}(\boldsymbol{o}_i, \pi_i) := f_i(\boldsymbol{o}_i, \boldsymbol{U}_{i,i-1}, \boldsymbol{U}_{i,i+1}, \pi_i)$ to obtain the equilibrium of player $i$, where $\boldsymbol{U}_{i,i-1}, \boldsymbol{U}_{i,i+1}$ are both utility matrices in $\mathcal{G}^0$.

### B.2.1 LOWERBOUNDS ON $(\epsilon, \delta)$-DP

For each player $i \in [N] \setminus \{1, 2\}$, assume the distributed algorithm may achieve the following guarantee.

$$\forall\, k \in \{0, 1\}, \quad \mathbb{E}\left[\left\|\pi_i^k - \pi_i^{*,k}\right\|^2\right] \leq \zeta_i \tag{Accuracy}$$

$$\forall\, \mathcal{C}_i \subseteq \Delta^{\mathcal{A}_i}, \quad \Pr(\pi_i^0 \in \mathcal{C}_i) \leq e^{\epsilon_i} \Pr(\pi_i^1 \in \mathcal{C}_i) + \delta_i, \tag{Privacy}$$

where $\pi_i^0, \pi_i^1$ is the output of the algorithm when deploying on player $i$ for $\mathcal{G}^0, \mathcal{G}^1$.

Let $\pi_i^{\mathrm{adv},k}$ be the random variable sampled from $\mathrm{adv}(\boldsymbol{o}_i^k, \cdot)$, where $\boldsymbol{o}_i^k$ is the observation in $\mathcal{G}^k$ with $k \in \{0, 1\}$. Since post-processing will not decrease the strength of DP, observing $\pi_i^{\mathrm{adv},k}$ instead of $\boldsymbol{o}_i^k$ will leak less information. Moreover, since the utility matrices of players $i > 2$ do not change, for any player $i > 2$ and $k \in \{0, 1\}$, we have

$$\mathbb{E}\left[\left\|\pi_i^k - \pi_i^{*,k}\right\|^2\right] = \mathbb{E}\left[\left\|\pi_i^{\mathrm{adv},k} - \pi_i^{*,k}\right\|^2\right].$$

Let the equilibrium of $\mathcal{G}^0$ be $\boldsymbol{\pi}^{*,0} := ((1,0),(0,1),\cdots)$ and the equilibrium of $\mathcal{G}^1$ be $\boldsymbol{\pi}^{*,1} := ((0,1),(1,0),\cdots)$. Then, by Markov inequality, for any $k \in \{0,1\}$ and $i > 2$, we have

$$\Pr\left(\left\|\pi_i^{\mathrm{adv},k} - \pi_i^{*,k}\right\|^2 \geq 1\right) \leq \frac{\mathbb{E}\left[\left\|\pi_i^{\mathrm{adv},k} - \pi_i^{*,k}\right\|^2\right]}{1} = \frac{\mathbb{E}\left[\left\|\pi_i^k - \pi_i^{*,k}\right\|^2\right]}{1} \leq \zeta_i.$$

Therefore, according to the privacy guarantee, for any $i > 2$, we have

$$1 - \zeta_i \leq \Pr\left(\left\|\pi_i^{\mathrm{adv},0} - \pi_i^{*,0}\right\|^2 \leq 1\right) \overset{(i)}{\leq} \Pr\left(\left\|\pi_i^{\mathrm{adv},0} - \pi_i^{*,1}\right\|^2 \geq 1\right)$$

$$\overset{(ii)}{\leq} e^{\epsilon_i} \Pr\left(\left\|\pi_i^{\mathrm{adv},1} - \pi_i^{*,1}\right\|^2 \geq 1\right) + \delta_i \leq e^{\epsilon_i} \zeta_i + \delta_i.$$

$(i)$ is because $\left\|\pi_i^{*,0} - \pi_i^{*,1}\right\|^2 = 2$. $(ii)$ is because post-processing will not decrease the strength of DP.

Therefore, we have $\zeta_i \geq \frac{1-\delta_i}{1+e^{\epsilon_i}}$ for $i > 2$. For $i \in \{1, 2\}$, we have $1 \geq \frac{1-\delta_i}{1+e^{\epsilon_i}}$ since $1 - \delta_i \leq 1$.

Moreover, we have

$$\frac{1}{N}\left(2 + \sum_{i=3}^N \zeta_i\right) \geq \frac{1}{N}\sum_{i=1}^N \frac{1-\delta_i}{1+e^{\epsilon_i}} \geq \frac{1}{N}\sum_{i=1}^N \frac{1-\max_{i\in[N]}\delta_i}{1+e^{\epsilon_i}} \overset{(i)}{\geq} \frac{1-\max_{i\in[N]}\delta_i}{1+\exp\left(\frac{1}{N}\sum_{i=1}^N \epsilon_i\right)}.$$

$(i)$ uses Jensen's inequality.

### B.2.2 LOWERBOUNDS ON $(\alpha, \epsilon)$-RÉNYI DP

By further post-processing on $\pi_i^{\mathrm{adv},k}$, we can get a random variable sampled from $\mu_i^k$, which is the distribution over $\{0,1\}$ indicating whether $\left\|\pi_i^{\mathrm{adv},k} - \pi_i^{*,0}\right\| \leq 1$. Then, for any $i > 2$, we have

$$D_\alpha\left(\mu_{\mathcal{G},i}, \mu_{\mathcal{G}',i}\right) := \epsilon_i \overset{(i)}{\geq} \mathrm{KL}(\mu_i^0, \mu_i^1) \overset{(ii)}{\geq} (1-\zeta_i)\log\frac{1-\zeta_i}{\zeta_i} + \zeta_i\log\frac{\zeta_i}{1-\zeta_i} = (1-2\zeta_i)\log\frac{1-\zeta_i}{\zeta_i}.$$

$(i)$ is because post-processing will not weaken Rényi DP (Mironov, 2017). $(ii)$ is because

$$\mu_i^0(0) = \Pr\left(\left\|\pi_i^{\text{adv},0} - \pi_i^{*,0}\right\| > 1\right) \leq \zeta_i$$

$$\mu_i^1(1) = \Pr\left(\left\|\pi_i^{\text{adv},1} - \pi_i^{*,0}\right\| \leq 1\right) \leq \Pr\left(\left\|\pi_i^{\text{adv},1} - \pi_i^{*,1}\right\| \geq 1\right) \leq \zeta_i.$$

Therefore, the KL-divergence is lowerbounded by $(1 - \zeta_i)\log\frac{1-\zeta_i}{\zeta_i} + \zeta_i\log\frac{\zeta_i}{1-\zeta_i}$.

When $\zeta_i \leq \frac{1}{4}$, we have $\epsilon_i \geq \frac{1}{2}\log\frac{3}{4\zeta_i}$. Therefore, $\zeta_i \geq \frac{3}{4e^{2\epsilon_i}}$. Moreover, if $\frac{1}{N}\sum_{i=1}^N \zeta_i \leq \frac{1}{16}$, by pigeon-hole principle, there exists a set of players $\mathcal{I} \subseteq [N] \setminus \{1, 2\}$ with $|\mathcal{I}| \geq \frac{N}{2}$, so that for any $i \in \mathcal{I}, \zeta_i \leq \frac{1}{4}$. This further implies that

$$\frac{1}{N}\sum_{i=1}^N \zeta_i \geq \frac{1}{2|\mathcal{I}|}\sum_{i\in\mathcal{I}} \zeta_i \qquad\qquad \frac{1}{N}\sum_{i=1}^N \epsilon_i \geq \frac{1}{2|\mathcal{I}|}\sum_{i\in\mathcal{I}} \epsilon_i,$$

since $\zeta_i, \epsilon_i \geq 0$ for any $i \in [N]$.

Therefore, finally,

$$\frac{1}{N}\sum_{i=1}^N \zeta_i \geq \frac{1}{2|\mathcal{I}|}\sum_{i\in\mathcal{I}} \zeta_i \geq \frac{3}{8|\mathcal{I}|}\sum_{i\in\mathcal{I}} e^{-2\epsilon_i} \stackrel{(i)}{\geq} \frac{3}{8}\exp\left(-\frac{2}{|\mathcal{I}|}\sum_{i\in\mathcal{I}}\epsilon_i\right) \geq \frac{3}{8}\exp\left(-\frac{4}{N}\sum_{i=1}^N \epsilon_i\right).$$

$(i)$ uses Jensen's inequality. $\qquad\qquad\qquad\qquad\qquad\qquad\qquad\qquad\qquad\qquad\qquad\qquad\qquad\square$

## B.3 OMITTED PROOF OF LEMMAS

**Lemma B.1.** *Consider the $2^N$ games $\left\{\mathcal{G}^{\mathcal{S}}\right\}_{\mathcal{S}\subseteq[N]}$ constructed above. There must exist two adjacent games $\mathcal{G}^{\mathcal{S}}$ and $\mathcal{G}^{\mathcal{S}\cup\{i\}}$ with $i \in [N] \setminus \mathcal{S}$, so that*

$$\mathbb{E}\left[\max_{\widehat{\pi}_{3i-1}\in\Delta^{\mathcal{A}_{3i-1}}}\left\langle\pi_{3i-1}^{\mathcal{S}} - \widehat{\pi}_{3i-1}, \boldsymbol{g}_{3i-1}^{\boldsymbol{\pi}^{\mathcal{S}}}(\mathcal{G}^{\mathcal{S}})\right\rangle + \max_{\widehat{\pi}_{3i}\in\Delta^{\mathcal{A}_{3i}}}\left\langle\pi_{3i}^{\mathcal{S}} - \widehat{\pi}_{3i}, \boldsymbol{g}_{3i}^{\boldsymbol{\pi}^{\mathcal{S}}}(\mathcal{G}^{\mathcal{S}})\right\rangle\right] \leq 7\zeta \qquad (18)$$

$$\mathbb{E}\left[\max_{\widehat{\pi}_{3i-1}\in\Delta^{\mathcal{A}_{3i-1}}}\left\langle\pi_{3i-1}^{\mathcal{S}\cup\{i\}} - \widehat{\pi}_{3i-1}, \boldsymbol{g}_{3i-1}^{\boldsymbol{\pi}^{\mathcal{S}\cup\{i\}}}(\mathcal{G}^{\mathcal{S}\cup\{i\}})\right\rangle + \max_{\widehat{\pi}_{3i}\in\Delta^{\mathcal{A}_{3i}}}\left\langle\pi_{3i}^{\mathcal{S}\cup\{i\}} - \widehat{\pi}_{3i}, \boldsymbol{g}_{3i}^{\boldsymbol{\pi}^{\mathcal{S}\cup\{i\}}}(\mathcal{G}^{\mathcal{S}\cup\{i\}})\right\rangle\right] \leq 7\zeta.$$
$$(19)$$

*Proof.* By the accuracy condition and non-negativity of $\max_{\widehat{\pi}_j\in\Delta^{\mathcal{A}_j}}\left\langle\pi_j^{\emptyset} - \widehat{\pi}_j, \boldsymbol{g}_j^{\boldsymbol{\pi}^{\emptyset}}(\mathcal{G}^{\emptyset})\right\rangle$, we have

$$\sum_{i\in[N]}\mathbb{E}\left[\max_{\widehat{\pi}_{3i-1}\in\Delta^{\mathcal{A}_{3i-1}}}\left\langle\pi_{3i-1}^{\emptyset} - \widehat{\pi}_{3i-1}, \boldsymbol{g}_{3i-1}^{\boldsymbol{\pi}^{\emptyset}}(\mathcal{G}^{\emptyset})\right\rangle + \max_{\widehat{\pi}_{3i}\in\Delta^{\mathcal{A}_{3i}}}\left\langle\pi_{3i}^{\emptyset} - \widehat{\pi}_{3i}, \boldsymbol{g}_{3i}^{\boldsymbol{\pi}^{\emptyset}}(\mathcal{G}^{\emptyset})\right\rangle\right] \leq 3N\zeta.$$

Therefore, by the pigeon-hole principle, there must exist a subset $\mathcal{I} \subseteq [N]$ with $|\mathcal{I}| \geq \lceil\frac{N}{2}\rceil + 1$, so that for any $i \in \mathcal{I}$, $\mathbb{E}\left[\max_{\widehat{\pi}_{3i-1}\in\Delta^{\mathcal{A}_{3i-1}}}\left\langle\pi_{3i-1}^{\emptyset} - \widehat{\pi}_{3i-1}, \boldsymbol{g}_{3i-1}^{\boldsymbol{\pi}^{\emptyset}}(\mathcal{G}^{\emptyset})\right\rangle + \max_{\widehat{\pi}_{3i}\in\Delta^{\mathcal{A}_{3i}}}\left\langle\pi_{3i}^{\emptyset} - \widehat{\pi}_{3i}, \boldsymbol{g}_{3i}^{\boldsymbol{\pi}^{\emptyset}}(\mathcal{G}^{\emptyset})\right\rangle\right] \leq 7\zeta$. The same guarantee holds for any $\mathcal{G}^{\{i\}}$ with $i \in [N]$.

Then, we will form a meta graph, with each node as a subset of $[N]$ so that there are $2^N$ nodes in total. For each two nodes $\mathcal{S}^1, \mathcal{S}^2$ with $|\mathcal{S}^1| \leq |\mathcal{S}^2|$, they are connected in the meta graph if and only if $\mathcal{G}^{\mathcal{S}^1}$ and $\mathcal{G}^{\mathcal{S}^2}$ are adjacent. In other words, $\mathcal{S}^2 = \mathcal{S}^1 \cup \{i\}$ and $i \in [N] \setminus \mathcal{S}^1$. The edge between $(\mathcal{S}^1, \mathcal{S}^2)$ is labeled as $i \in [N]$ if $\mathcal{S}^2 \setminus \mathcal{S}^1 = \{i\}$. Therefore, each node has $N$ edges and their labels are different from each other. Then, there are $2^N$ nodes in the meta graph and $N2^{N-1}$ edges.

For each node $\mathcal{S} \subseteq [N]$, the set $\mathcal{I}$ constructed above can be viewed as selecting edges with labels in $\mathcal{I}$. Then, for each node $\mathcal{S}$ and its selected edge with label $i$, it is guaranteed that

$$\mathbb{E}\left[\max_{\widehat{\pi}_{3i-1}\in\Delta^{\mathcal{A}_{3i-1}}}\left\langle\pi_{3i-1}^{\mathcal{S}} - \widehat{\pi}_{3i-1}, \boldsymbol{g}_{3i-1}^{\boldsymbol{\pi}^{\mathcal{S}}}(\mathcal{G}^{\mathcal{S}})\right\rangle + \max_{\widehat{\pi}_{3i}\in\Delta^{\mathcal{A}_{3i}}}\left\langle\pi_{3i}^{\mathcal{S}} - \widehat{\pi}_{3i}, \boldsymbol{g}_{3i}^{\boldsymbol{\pi}^{\mathcal{S}}}(\mathcal{G}^{\mathcal{S}})\right\rangle\right] \leq 7\zeta. \quad (21)$$

Because $|\mathcal{I}| \geq \left\lceil \frac{N}{2} \right\rceil + 1$ and there are $N2^{N-1}$ edges in total, by pigeon-hole principle, there must exist an edge being selected twice. In other words, there exist two adjacent nodes $\mathcal{S}^1, \mathcal{S}^2$ that both select edge with label $i$. By definition, assume $|\mathcal{S}^1| \leq |\mathcal{S}^2|$ without loss of generality, then $\mathcal{S}^2 = \mathcal{S}^1 \cup \{i\}$. Therefore, (21) is satisfied for both $\mathcal{S}^1, \mathcal{S}^2$ on $i$. $\qquad\square$

## C  PROOF OF THEOREM 5.1

**Theorem 5.1.** *Consider Algorithm 1. Let $A = \max_{i \in [N]} |\mathcal{A}_i|$. The update-rule can achieve the following guarantee in any polymatrix game. For any $T > 0$, player $i \in [N]$ and strategy $\pi_i \in \Delta^{\mathcal{A}_i}$,*

$$
\frac{1}{NT} \sum_{i=1}^{N} \sum_{t=1}^{T} \mathbb{E}\left[\left\langle \boldsymbol{g}_i^{\boldsymbol{\pi}^{(t+1)}}, \pi_i^{(t+1)} - \pi_i \right\rangle\right]
$$
$$
\leq \frac{1}{\eta T} + A\frac{\sigma^2}{2\eta} + \left(\frac{2\eta^2}{\sigma} + \frac{7\sigma}{2}\right) A^{\frac{3}{2}} + \frac{1}{2\left(\overline{\mathcal{N}}\right)^{4/9} \log N} + \frac{2\eta\sqrt{A}}{\sigma\left(\overline{\mathcal{N}}\right)^{4/9} \log N}. \tag{10}
$$

We will give a proof sketch of Theorem 5.1 first.

### C.1  PROOF SKETCH

To show the convergence of Algorithm 1, by Lemma D.4. in Liu et al. (2025), we have

$$
\frac{\tau_i}{2} \left\| \pi_i^{(t+1)} \right\|^2 - \frac{\tau_i}{2} \|\pi_i\|^2 + \left\langle \bar{\boldsymbol{g}}_i^{(t)}, \pi_i^{(t+1)} - \pi_i \right\rangle
$$
$$
\leq \frac{1}{2\eta} \left\| \pi_i - \bar{\pi}_i^{(t)} \right\|^2 - \frac{1 + \eta\tau_i}{2\eta} \left\| \pi_i - \pi_i^{(t+1)} \right\|^2 - \frac{1}{2\eta} \left\| \pi_i^{(t+1)} - \bar{\pi}_i^{(t)} \right\|^2.
$$

Moreover, $\left| \frac{\tau_i}{2} \left\| \pi_i^{(t+1)} \right\|^2 - \frac{\tau_i}{2} \|\pi_i\|^2 \right| \leq \frac{\tau_i}{2}$. Then, by adding $\left\langle \boldsymbol{g}_i^{\boldsymbol{\pi}^{(t+1)}} - \bar{\boldsymbol{g}}_i^{(t)}, \pi_i^{(t+1)} - \pi_i \right\rangle$ on both sides, we have

$$
\left\langle \boldsymbol{g}_i^{\boldsymbol{\pi}^{(t+1)}}, \pi_i^{(t+1)} - \pi_i \right\rangle - \frac{\tau_i}{2}
$$
$$
\leq \frac{1}{2\eta} \left\| \pi_i - \bar{\pi}_i^{(t)} \right\|^2 - \frac{1 + \eta\tau_i}{2\eta} \left\| \pi_i - \pi_i^{(t+1)} \right\|^2
$$
$$
- \frac{1}{2\eta} \left\| \pi_i^{(t+1)} - \bar{\pi}_i^{(t)} \right\|^2 + \left\langle \boldsymbol{g}_i^{\boldsymbol{\pi}^{(t+1)}} - \bar{\boldsymbol{g}}_i^{(t)}, \pi_i^{(t+1)} - \pi_i \right\rangle.
$$

Moreover, $\mathbb{E}\left[\left\langle \boldsymbol{g}_i^{\boldsymbol{\pi}^{(t+1)}} - \bar{\boldsymbol{g}}_i^{(t)}, \pi_i^{(t+1)} - \pi_i \right\rangle\right] \leq \left(\frac{\eta^2}{\sigma} + \frac{7\sigma}{2}\right) A^{\frac{3}{2}} + \frac{2\eta\sqrt{A}}{|\mathcal{N}(i)|\sigma} \sum_{j \in \mathcal{N}(i)} \tau_j$, where the complete proof is postponed to Lemma C.1. Therefore, by taking expectation on both sides and telescoping, we have

$$
\sum_{t=1}^{T} \mathbb{E}\left[\left\langle \boldsymbol{g}_i^{\boldsymbol{\pi}^{(t+1)}}, \pi_i^{(t+1)} - \pi_i \right\rangle\right]
$$
$$
\leq \frac{1}{2\eta} \mathbb{E}\left[\left\| \pi_i - \bar{\pi}_i^{(1)} \right\|^2\right] + \left(\frac{\eta^2}{\sigma} + \frac{7\sigma}{2}\right) A^{\frac{3}{2}} T + \frac{\tau_i}{2} T + \sum_{t=2}^{T} \mathbb{E}\left[\left\| \pi_i - \bar{\pi}_i^{(t)} \right\|^2 - \left\| \pi_i - \pi_i^{(t)} \right\|^2\right]
$$
$$
+ \frac{2\eta\sqrt{A}}{|\mathcal{N}(i)|\sigma} \sum_{j \in \mathcal{N}(i)} \tau_j.
$$

Further, $\mathbb{E}\left[\left\|\pi_i - \bar{\pi}_i^{(t)}\right\|^2 - \left\|\pi_i - \pi_i^{(t)}\right\|^2\right]$ is bounded by,

$$\mathbb{E}\left[\left\|\pi_i - \mathrm{Proj}_{\Delta^{\mathcal{A}_i}}\left(\pi_i^{(t)} + \boldsymbol{n}_i^{(t)}\right)\right\|^2 - \left\|\pi_i - \pi_i^{(t)}\right\|^2\right]$$

$$\leq \mathbb{E}\left[\left\|\pi_i - \pi_i^{(t)} - \boldsymbol{n}_i^{(t)}\right\|^2 - \left\|\pi_i - \pi_i^{(t)}\right\|^2\right]$$

$$= \mathbb{E}\left[2\left\langle \boldsymbol{n}_i^{(t)}, \pi_i - \pi_i^{(t)}\right\rangle + \left\|\boldsymbol{n}_i^{(t)}\right\|^2\right]$$

$$\overset{(i)}{=} \mathbb{E}\left[\left\|\boldsymbol{n}_i^{(t)}\right\|^2\right] = A\sigma^2.$$

$(i)$ is because $\mathbb{E}\left[\boldsymbol{n}_i^{(t)}\right] = \boldsymbol{0}$ and $\boldsymbol{n}_i^{(t)}$ is independent of $\pi_i - \pi_i^{(t)}$. By aggregating the results above and substitute $\tau_i$ by its value in Algorithm 1, the proof is concluded.

## C.2 FORMAL PROOF OF THEOREM 5.1

By Lemma D.4. in Liu et al. (2025), since Algorithm 1 is equivalent to $\pi_i^{(t+1)} = \arg\min_{\pi_i \in \Delta^{\mathcal{A}_i}} \left\langle \pi_i, \bar{\boldsymbol{g}}_i^{(t)}\right\rangle + \frac{\tau_i}{2}\|\pi_i\|^2 + \frac{1}{2\eta}\left\|\pi_i - \bar{\pi}_i^{(t)}\right\|^2$, for any $\pi_i \in \Delta^{\mathcal{A}_i}$, we have

$$\frac{\eta\tau_i}{2}\left\|\pi_i^{(t+1)}\right\|^2 - \frac{\eta\tau_i}{2}\|\pi_i\|^2 + \eta\left\langle \bar{\boldsymbol{g}}_i^{(t)}, \pi_i^{(t+1)} - \pi_i\right\rangle$$

$$\leq \frac{1}{2}\left\|\pi_i - \bar{\pi}_i^{(t)}\right\|^2 - \frac{1+\eta\tau_i}{2}\left\|\pi_i - \pi_i^{(t+1)}\right\|^2 - \frac{1}{2}\left\|\pi_i^{(t+1)} - \bar{\pi}_i^{(t)}\right\|^2.$$

Then, by adding $\eta\left\langle \bar{\boldsymbol{g}}_i^{(t+1)} - \bar{\boldsymbol{g}}_i^{(t)}, \pi_i^{(t+1)} - \pi_i\right\rangle$ on both sides, we have

$$\frac{\eta\tau_i}{2}\left\|\pi_i^{(t+1)}\right\|^2 - \frac{\eta\tau_i}{2}\|\pi_i\|^2 + \eta\left\langle \bar{\boldsymbol{g}}_i^{(t+1)}, \pi_i^{(t+1)} - \pi_i\right\rangle \tag{22}$$

$$\leq \frac{1}{2}\left\|\pi_i - \bar{\pi}_i^{(t)}\right\|^2 - \frac{1+\eta\tau_i}{2}\left\|\pi_i - \pi_i^{(t+1)}\right\|^2 - \frac{1}{2}\left\|\pi_i^{(t+1)} - \bar{\pi}_i^{(t)}\right\|^2 + \eta\left\langle \bar{\boldsymbol{g}}_i^{(t+1)} - \bar{\boldsymbol{g}}_i^{(t)}, \pi_i^{(t+1)} - \pi_i\right\rangle.$$

Moreover,

$$\left\langle \bar{\boldsymbol{g}}_i^{(t+1)}, \pi_i^{(t+1)} - \pi_i\right\rangle = \left\langle \boldsymbol{g}_i^{\boldsymbol{\pi}^{(t+1)}}, \pi_i^{(t+1)} - \pi_i\right\rangle + \left\langle \bar{\boldsymbol{g}}_i^{(t+1)} - \boldsymbol{g}_i^{\boldsymbol{\pi}^{(t+1)}}, \pi_i^{(t+1)} - \pi_i\right\rangle$$

$$\geq \left\langle \boldsymbol{g}_i^{\boldsymbol{\pi}^{(t+1)}}, \pi_i^{(t+1)} - \pi_i\right\rangle - \left\|\bar{\boldsymbol{g}}_i^{(t+1)} - \boldsymbol{g}_i^{\boldsymbol{\pi}^{(t+1)}}\right\|_\infty \cdot \left\|\pi_i^{(t+1)} - \pi_i\right\|_1$$

$$\geq \left\langle \boldsymbol{g}_i^{\boldsymbol{\pi}^{(t+1)}}, \pi_i^{(t+1)} - \pi_i\right\rangle - 2\left\|\frac{1}{|\mathcal{N}(i)|}\sum_{j\in\mathcal{N}(i)}\boldsymbol{U}_{i,j}\left(\pi_j^{(t+1)} - \bar{\pi}_j^{(t+1)}\right)\right\|_\infty$$

$$\overset{(i)}{\geq} \left\langle \boldsymbol{g}_i^{\boldsymbol{\pi}^{(t+1)}}, \pi_i^{(t+1)} - \pi_i\right\rangle - \frac{2}{|\mathcal{N}(i)|}\sum_{j\in\mathcal{N}(i)}\left\|\pi_j^{(t+1)} - \bar{\pi}_j^{(t+1)}\right\|_1.$$

$(i)$ is because $\boldsymbol{U}_{i,j} \in [-1,1]^{\mathcal{A}_i \times \mathcal{A}_j}$. Recall that $A := \max_{i\in[N]}|\mathcal{A}_i|$, we have

$$\left\|\pi_j^{(t+1)} - \bar{\pi}_j^{(t+1)}\right\|_1 \leq \sqrt{A}\left\|\pi_j^{(t+1)} - \bar{\pi}_j^{(t+1)}\right\| = \sqrt{A}\left\|\pi_j^{(t+1)} - \mathrm{Proj}_{\Delta^{\mathcal{A}_i}}\left(\pi_j^{(t+1)} + \boldsymbol{n}_j^{(t+1)}\right)\right\|$$

$$\leq \sqrt{A}\left\|\boldsymbol{n}_j^{(t+1)}\right\|.$$

By taking the expectation on both sides of (22), we have,

$$
\mathbb{E}\left[\left(\frac{\eta\tau_i}{2}\left\|\pi_i^{(t+1)}\right\|^2 - \frac{\eta\tau_i}{2}\left\|\pi_i\right\|^2 + \eta\left\langle -\frac{1}{|\mathcal{N}(i)|}\sum_{j\in\mathcal{N}(i)}\boldsymbol{U}_{i,j}\pi_j^{(t+1)}, \pi_i^{(t+1)} - \pi_i\right\rangle\right)\right] \quad (23)
$$

$$
\leq \mathbb{E}\left[\frac{1}{2}\left\|\pi_i - \bar{\pi}_i^{(t)}\right\|^2 - \frac{1+\eta\tau_i}{2}\left\|\pi_i - \pi_i^{(t+1)}\right\|^2 - \frac{1}{2}\left\|\pi_i^{(t+1)} - \bar{\pi}_i^{(t)}\right\|^2\right]
$$

$$
+ \eta\mathbb{E}\left[\left\langle \bar{\boldsymbol{g}}_i^{(t+1)} - \bar{\boldsymbol{g}}_i^{(t)}, \pi_i^{(t+1)} - \pi_i\right\rangle + \frac{2\sqrt{A}}{|\mathcal{N}(i)|}\sum_{j\in\mathcal{N}(i)}\left\|\boldsymbol{n}_j^{(t+1)}\right\|\right].
$$

By Jensen's Inequality, $\mathbb{E}\left[\left\|\boldsymbol{n}_j^{(t+1)}\right\|\right] \leq \sqrt{\mathbb{E}\left[\left\|\boldsymbol{n}_j^{(t+1)}\right\|^2\right]} = \sqrt{|\mathcal{A}_j|}\sigma$. Therefore,

$$
\frac{2\sqrt{A}}{|\mathcal{N}(i)|}\sum_{j\in\mathcal{N}(i)}\mathbb{E}\left[\left\|\boldsymbol{n}_j^{(t+1)}\right\|\right] \leq 2A\sigma.
$$

Moreover, since $\|\pi_i\|^2 \in [\frac{1}{|\mathcal{A}_i|}, 1]$, we have

$$
\mathbb{E}\left[\eta\left\langle -\frac{1}{|\mathcal{N}(i)|}\sum_{j\in\mathcal{N}(i)}\boldsymbol{U}_{i,j}\pi_j^{(t+1)}, \pi_i^{(t+1)} - \pi_i\right\rangle\right]
$$

$$
\leq \mathbb{E}\left[\frac{1}{2}\left\|\pi_i - \bar{\pi}_i^{(t)}\right\|^2 - \frac{1+\eta\tau_i}{2}\left\|\pi_i - \pi_i^{(t+1)}\right\|^2 - \frac{1}{2}\left\|\pi_i^{(t+1)} - \bar{\pi}_i^{(t)}\right\|^2 + \eta\left\langle \bar{\boldsymbol{g}}_i^{(t+1)} - \bar{\boldsymbol{g}}_i^{(t)}, \pi_i^{(t+1)} - \pi_i\right\rangle\right]
$$

$$
+ 2A\eta\sigma + \frac{\eta}{2}\tau_i.
$$

To bound $\mathbb{E}\left[\left\langle \bar{\boldsymbol{g}}_i^{(t+1)} - \bar{\boldsymbol{g}}_i^{(t)}, \pi_i^{(t+1)} - \pi_i\right\rangle\right]$, we have the following lemma.

**Lemma C.1.** *Consider Algorithm 1. For any player $i \in [N]$, we have*

$$
\mathbb{E}\left[\left\langle \bar{\boldsymbol{g}}_i^{(t+1)} - \bar{\boldsymbol{g}}_i^{(t)}, \pi_i^{(t+1)} - \pi_i\right\rangle\right] \leq \left(\frac{2\eta^2}{\sigma} + \frac{3\sigma}{2}\right)A^{\frac{3}{2}} + \frac{2\eta\sqrt{A}}{|\mathcal{N}(i)|\sigma}\sum_{j\in\mathcal{N}(i)}\tau_j. \quad (24)
$$

The proof is postponed to the end of this section.

We can further bound $\mathbb{E}\left[\left\|\pi_i - \bar{\pi}_i^{(t)}\right\|^2 - \left\|\pi_i - \pi_i^{(t)}\right\|^2\right]$ by

$$
\mathbb{E}\left[\left\|\pi_i - \mathrm{Proj}_{\Delta^{\mathcal{A}_i}}\left(\pi_i^{(t)} + \boldsymbol{n}_i^{(t)}\right)\right\|^2 - \left\|\pi_i - \pi_i^{(t)}\right\|^2\right] \leq \mathbb{E}\left[\left\|\pi_i - \pi_i^{(t)} - \boldsymbol{n}_i^{(t)}\right\|^2 - \left\|\pi_i - \pi_i^{(t)}\right\|^2\right]
$$

$$
= \mathbb{E}\left[2\left\langle \boldsymbol{n}_i^{(t)}, \pi_i^{(t)} - \pi_i\right\rangle + \left\|\boldsymbol{n}_i^{(t)}\right\|^2\right]
$$

$$
\overset{(i)}{=} \mathbb{E}\left[\left\|\boldsymbol{n}_i^{(t)}\right\|^2\right] = A\sigma^2.
$$

$(i)$ is because $\mathbb{E}\left[\boldsymbol{n}_i^{(t)}\right] = \mathbf{0}$ and $\boldsymbol{n}_i^{(t)}$ is independent of $\pi_i - \pi_i^{(t)}$.

Finally, by telescoping,

$$
\begin{aligned}
&\sum_{t=1}^{T} \mathbb{E}\left[\left\langle -\frac{1}{|\mathcal{N}(i)|} \sum_{j \in \mathcal{N}(i)} \boldsymbol{U}_{i,j} \pi_j^{(t+1)}, \pi_i^{(t+1)} - \pi_i \right\rangle\right] \\
&\leq \frac{1}{2\eta}\left\| \pi_1 - \bar{\pi}_1^{(1)} \right\|^2 + A\frac{\sigma^2}{2\eta}T + \left(\frac{2\eta^2}{\sigma} + \frac{3\sigma}{2}\right) A^{\frac{3}{2}}T + 2A\sigma T + \frac{1}{2}\tau_i T + \frac{2\eta\sqrt{A}}{|\mathcal{N}(i)|\,\sigma} \sum_{j \in \mathcal{N}(i)} \tau_j T \\
&\leq \frac{1}{\eta} + A\frac{\sigma^2}{2\eta}T + \left(\frac{2\eta^2}{\sigma} + \frac{7\sigma}{2}\right) A^{\frac{3}{2}}T + \frac{1}{2}\tau_i T + \frac{2\eta\sqrt{A}}{|\mathcal{N}(i)|\,\sigma} \sum_{j \in \mathcal{N}(i)} \tau_j T.
\end{aligned}
\tag{25}
$$

Further, by taking summation over all player $i \in [N]$ and averaging, we have

$$
\begin{aligned}
&\frac{1}{N}\sum_{i=1}^{N}\sum_{t=1}^{T} \mathbb{E}\left[\left\langle -\frac{1}{|\mathcal{N}(i)|} \sum_{j \in \mathcal{N}(i)} \boldsymbol{U}_{i,j} \pi_j^{(t+1)}, \pi_i^{(t+1)} - \pi_i \right\rangle\right] \\
&\leq \frac{1}{\eta} + A\frac{\sigma^2}{2\eta}T + \left(\frac{2\eta^2}{\sigma} + \frac{7\sigma}{2}\right) A^{\frac{3}{2}}T + \frac{T}{2N}\sum_{i=1}^{N}\tau_i + \frac{1}{N}\sum_{i=1}^{N}\frac{2\eta\sqrt{A}}{|\mathcal{N}(i)|\,\sigma} \sum_{j \in \mathcal{N}(i)} \tau_j T.
\end{aligned}
$$

Then, we can further bound the summation over $\tau_i$ by Lemma C.2.

**Lemma C.2.** *When $\tau_i = \frac{c}{|\mathcal{N}(i)|}$ for any $i \in [N]$, where $c > 0$ is a game-dependent constant, we have*

$$
\frac{1}{N}\sum_{i=1}^{N}\tau_i = \frac{c}{\overline{N}} \qquad\qquad \frac{1}{N}\sum_{i=1}^{N}\frac{1}{|\mathcal{N}(i)|}\sum_{j\in\mathcal{N}(i)}\tau_j \leq \frac{c}{\overline{N}}.
\tag{26}
$$

The proof is postponed to the end of this section. Therefore, since $c = \frac{\left(\overline{N}\right)^{5/9}}{\log N}$ in Algorithm 1, we have

$$
\begin{aligned}
&\frac{1}{N}\sum_{i=1}^{N}\frac{1}{T}\sum_{t=1}^{T} \mathbb{E}\left[\left\langle -\frac{1}{|\mathcal{N}(i)|} \sum_{j \in \mathcal{N}(i)} \boldsymbol{U}_{i,j} \pi_j^{(t+1)}, \pi_i^{(t+1)} - \pi_i \right\rangle\right] \\
&\leq \frac{1}{\eta T} + A\frac{\sigma^2}{2\eta} + \left(\frac{2\eta^2}{\sigma} + \frac{7\sigma}{2}\right) A^{\frac{3}{2}} + \frac{1}{2\left(\overline{N}\right)^{4/9}\log N} + \frac{2\eta\sqrt{A}}{\sigma\left(\overline{N}\right)^{4/9}\log N}. \qquad\qquad \square
\end{aligned}
$$

## C.3 Omitted Proof of Appendix C

**Lemma C.1.** *Consider Algorithm 1. For any player $i \in [N]$, we have*

$$
\mathbb{E}\left[\left\langle \bar{\boldsymbol{g}}_i^{(t+1)} - \bar{\boldsymbol{g}}_i^{(t)}, \pi_i^{(t+1)} - \pi_i \right\rangle\right] \leq \left(\frac{2\eta^2}{\sigma} + \frac{3\sigma}{2}\right) A^{\frac{3}{2}} + \frac{2\eta\sqrt{A}}{|\mathcal{N}(i)|\,\sigma}\sum_{j\in\mathcal{N}(i)}\tau_j.
\tag{24}
$$

*Proof.* By Hölder's Inequality, we have

$$
\mathbb{E}\left[\left\langle \bar{\boldsymbol{g}}_i^{(t+1)} - \bar{\boldsymbol{g}}_i^{(t)}, \pi_i^{(t+1)} - \pi_i \right\rangle\right] \leq \mathbb{E}\left[\left\| \bar{\boldsymbol{g}}_i^{(t+1)} - \bar{\boldsymbol{g}}_i^{(t)} \right\| \cdot \left\| \pi_i^{(t+1)} - \pi_i \right\|\right].
$$

Moreover,

$$
\begin{aligned}
\left\|\bar{\boldsymbol{g}}_i^{(t+1)} - \bar{\boldsymbol{g}}_i^{(t)}\right\| &= \frac{1}{|\mathcal{N}(i)|} \left\| \sum_{j \in \mathcal{N}(i)} \boldsymbol{U}_{i,j} \left( \mathrm{Proj}_{\Delta^{\mathcal{A}_j}} \left( \pi_j^{(t+1)} + \boldsymbol{n}_j^{(t+1)} \right) - \bar{\pi}_j^{(t)} \right) \right\| \\
&\leq \frac{\sqrt{|\mathcal{A}_i|}}{|\mathcal{N}(i)|} \sum_{j \in \mathcal{N}(i)} \left\| \mathrm{Proj}_{\Delta^{\mathcal{A}_j}} \left( \pi_j^{(t+1)} + \boldsymbol{n}_j^{(t+1)} \right) - \bar{\pi}_j^{(t)} \right\|_1 \\
&\leq \frac{\sqrt{|\mathcal{A}_i|}}{|\mathcal{N}(i)|} \sum_{j \in \mathcal{N}(i)} \sqrt{|\mathcal{A}_j|} \left\| \mathrm{Proj}_{\Delta^{\mathcal{A}_j}} \left( \pi_j^{(t+1)} + \boldsymbol{n}_j^{(t+1)} \right) - \bar{\pi}_j^{(t)} \right\| \\
&\leq \frac{\sqrt{|\mathcal{A}_i|}}{|\mathcal{N}(i)|} \sum_{j \in \mathcal{N}(i)} \sqrt{|\mathcal{A}_j|} \left\| \pi_j^{(t+1)} + \boldsymbol{n}_j^{(t+1)} - \bar{\pi}_j^{(t)} \right\|.
\end{aligned}
$$

Therefore,

$$
\begin{aligned}
&\mathbb{E}\left[ \left\langle \bar{\boldsymbol{g}}_i^{(t+1)} - \bar{\boldsymbol{g}}_i^{(t)}, \pi_i^{(t+1)} - \pi_i \right\rangle \right] \\
&\leq \frac{A}{|\mathcal{N}(i)|} \sum_{j \in \mathcal{N}(i)} \mathbb{E}\left[ \left\| \pi_j^{(t+1)} + \boldsymbol{n}_j^{(t+1)} - \bar{\pi}_j^{(t)} \right\| \cdot \left\| \pi_i^{(t+1)} - \pi_i \right\| \right] \\
&\leq \frac{A}{|\mathcal{N}(i)|} \sum_{j \in \mathcal{N}(i)} \left( \frac{1}{\sigma\sqrt{A}} \mathbb{E}\left[ \left\| \pi_j^{(t+1)} + \boldsymbol{n}_j^{(t+1)} - \bar{\pi}_j^{(t)} \right\|^2 \right] + \frac{\sigma\sqrt{A}}{4} \mathbb{E}\left[ \left\| \pi_i^{(t+1)} - \pi_i \right\|^2 \right] \right).
\end{aligned}
$$

Furthermore, since $\mathbb{E}\left[ \boldsymbol{n}_j^{(t+1)} \right] = 0$ and $\boldsymbol{n}_j^{(t+1)}$ is independent of $\pi_j^{(t+1)}, \bar{\pi}_j^{(t)}$, we have

$$
\begin{aligned}
\mathbb{E}\left[ \left\| \pi_j^{(t+1)} + \boldsymbol{n}_j^{(t+1)} - \bar{\pi}_j^{(t)} \right\|^2 \right] &= \mathbb{E}\left[ \left\| \pi_j^{(t+1)} - \bar{\pi}_j^{(t)} \right\|^2 \right] + \mathbb{E}\left[ \left\| \boldsymbol{n}_j^{(t+1)} \right\|^2 \right] \\
&\leq \mathbb{E}\left[ \left\| \mathrm{Proj}_{\Delta^{\mathcal{A}_j}} \left( \frac{\bar{\pi}_j^{(t)} - \eta \boldsymbol{g}_j^{(t)}}{1 + \eta\tau_j} \right) - \bar{\pi}_j^{(t)} \right\|^2 \right] + A\sigma^2 \\
&\leq \mathbb{E}\left[ \left\| \frac{\bar{\pi}_j^{(t)} - \eta \boldsymbol{g}_j^{(t)}}{1 + \eta\tau_j} - \bar{\pi}_j^{(t)} \right\|^2 \right] + A\sigma^2 \\
&= \mathbb{E}\left[ \left\| \frac{\eta\tau_j}{1 + \eta\tau_j} \bar{\pi}_j^{(t)} - \frac{\eta}{1 + \eta\tau_j} \boldsymbol{g}_j^{(t)} \right\|^2 \right] + A\sigma^2 \\
&\overset{(i)}{\leq} 2\eta\tau_j \mathbb{E}\left[ \left\| \bar{\pi}_j^{(t)} \right\|^2 \right] + 2\eta^2 \mathbb{E}\left[ \left\| \boldsymbol{g}_j^{(t)} \right\|^2 \right] + A\sigma^2 \\
&\overset{(ii)}{\leq} 2\eta\tau_j + 2\eta^2 A + A\sigma^2.
\end{aligned}
$$

$(i)$ is because $(a+b)^2 \leq 2a^2 + 2b^2$ for any $a, b \in \mathbb{R}$ and $1 + \eta\tau_j \geq \max\{1, \eta\tau_j\}$. $(ii)$ is because $\boldsymbol{U}_{i,j} \in [-1, 1]^{\mathcal{A}_i \times \mathcal{A}_j}$. Also, $\frac{\sigma\sqrt{A}}{4} \mathbb{E}\left[ \left\| \pi_i^{(t+1)} - \pi_i \right\|^2 \right] \leq \frac{\sigma\sqrt{A}}{2}$. Therefore,

$$
\mathbb{E}\left[ \left\langle \bar{\boldsymbol{g}}_i^{(t+1)} - \bar{\boldsymbol{g}}_i^{(t)}, \pi_i^{(t+1)} - \pi_i \right\rangle \right] \leq \left( \frac{2\eta^2}{\sigma} + \frac{3\sigma}{2} \right) A^{\frac{3}{2}} + \frac{2\eta\sqrt{A}}{|\mathcal{N}(i)|\sigma} \sum_{j \in \mathcal{N}(i)} \tau_j. \qquad \square
$$

**Lemma C.2.** *When $\tau_i = \frac{c}{|\mathcal{N}(i)|}$ for any $i \in [N]$, where $c > 0$ is a game-dependent constant, we have*

$$
\frac{1}{N} \sum_{i=1}^{N} \tau_i = \frac{c}{\mathcal{N}} \qquad\qquad \frac{1}{N} \sum_{i=1}^{N} \frac{1}{|\mathcal{N}(i)|} \sum_{j \in \mathcal{N}(i)} \tau_j \leq \frac{c}{\mathcal{N}}. \qquad (26)
$$

*Proof.* By definition,

$$\frac{1}{N} \sum_{i=1}^{N} \tau_i = \frac{c}{N} \sum_{i=1}^{N} \frac{1}{|\mathcal{N}(i)|} = \frac{c}{\mathcal{N}}.$$

Moreover,

$$\frac{1}{N} \sum_{i=1}^{N} \frac{1}{|\mathcal{N}(i)|} \sum_{j \in \mathcal{N}(i)} \tau_j = \frac{c}{N} \sum_{i=1}^{N} \sum_{j \in \mathcal{N}(i)} \frac{1}{|\mathcal{N}(i)| \cdot |\mathcal{N}(j)|}$$

$$\leq \frac{c}{N} \sum_{i=1}^{N} \sum_{j \in \mathcal{N}(i)} \left( \frac{1}{2|\mathcal{N}(i)|^2} + \frac{1}{2|\mathcal{N}(j)|^2} \right)$$

$$\overset{(i)}{=} \frac{c}{N} \sum_{i=1}^{N} \frac{2|\mathcal{N}(i)|}{2|\mathcal{N}(i)|^2}$$

$$= \frac{c}{\mathcal{N}}.$$

$(i)$ is because each player contributes $\frac{1}{2|\mathcal{N}(i)|^2}$ to the summation 2 times on each edge, *i.e.* $2|\mathcal{N}(i)|$ times in total. $\qquad\square$

## D    PROOF OF SECTION 6

**Theorem 6.1.** *Consider Algorithm 1 and any two adjacent polymatrix games $\mathcal{G}, \mathcal{G}'$ that differ at $(v_1, v_2)$. Let $dist(i, j)$ be the length of the shortest path between players $i, j$, which is $\infty$ when $i, j$ are not connected. The update-rule guarantees the following for any $T > 0$ and player $i \in [N]$,*

$$\frac{1}{N} \sum_{i=1}^{N} D_\alpha \left( \mu_{\mathcal{G},i}^{(T)}, \mu_{\mathcal{G}',i}^{(T)} \right) \leq \frac{\alpha \eta^2}{\sigma^2} \min \{\clubsuit, \spadesuit\} T, \tag{11}$$

*where*

$$\clubsuit := \frac{16 A^3 (\log N)^2}{(\mathcal{N})^{4/9}} + \frac{4A}{N} \tag{12}$$

$$\spadesuit := \frac{2A}{N} \sum_{i=1}^{N} \mathbb{1} \left( T > \min \{dist(i, v_1), dist(i, v_2)\} \right). \tag{13}$$

We will give a proof sketch of Theorem 6.1 first.

### D.1    PROOF SKETCH: THE CASE OF DENSE GRAPHS ($\clubsuit$)

Firstly, we will introduce the chain rule of Rényi divergence.

**Lemma D.1** (Chain Rule of Rényi divergence). *For any distribution $p, q$ over random variables $X^1, X^2$, for any $\alpha \geq 1$, we have*

$$D_\alpha \left( p(X^1, X^2), q(X^1, X^2) \right)$$
$$\leq D_\alpha \left( p(X^1), q(X^1) \right) + \sup_{\widehat{x}^1} D_\alpha \left( p(X^2 \mid X^1 = \widehat{x}^1), q(X^2 \mid X^1 = \widehat{x}^1) \right). \tag{27}$$

A proof of the above lemma is provided in Appendix D.5 for completeness. By Lemma D.1, the divergence between the distribution over all observations can be decomposed into the divergence between the distribution over the observation at a single timestep. Formally, for any $\alpha \geq 1$, adjacent games $\mathcal{G} \sim \mathcal{G}'$, and player $i \in [N]$,

$$D_\alpha \left( \mu_{\mathcal{G},i}^{(T)}, \mu_{\mathcal{G}',i}^{(T)} \right)$$
$$\leq \sum_{t=1}^{T} \sup_{\left( \widehat{o}^{(1)}, \cdots, \widehat{o}^{(t-1)} \right)} D_\alpha \left( \mu_{\mathcal{G},i}^{(t)}(\cdot \mid \widehat{o}^{(1)}, \cdots, \widehat{o}^{(t-1)}), \mu_{\mathcal{G}',i}^{(t)}(\cdot \mid \widehat{o}^{(1)}, \cdots, \widehat{o}^{(t-1)}) \right).$$

Note that the observation here $\widehat{o}^{(s)} = \pi_i^{(s)} + \boldsymbol{n}_i^{(s)}$. When the graph is dense, *i.e.* $\overline{\mathcal{N}} = N^p$ for some $p > 0$, the degree $|\mathcal{N}(i)|$ of most players will close to $\overline{\mathcal{N}}$.

Then, since post-processing will not increase the privacy budget, we can augment the set of observations to bound $D_\alpha\left(\mu_{\mathcal{G},i}^{(T)}, \mu_{\mathcal{G}',i}^{(T)}\right)$. Specifically, for two adjacent games $\mathcal{G} \sim \mathcal{G}'$ that differs at the utility matrices on edge $(v_1, v_2)$, the augmented observation will include $\left\{\pi_j^{(t)} + \boldsymbol{n}_j^{(t)}\right\}_{j \in \mathcal{S}}$, where $\mathcal{S} \supseteq [N] \setminus \{v_1, v_2\}$ and $v_1, v_2$ will be included in $\mathcal{S}$ if their degrees are no less than $\overline{\mathcal{N}}^{2/9}$. Let $\mu_{\mathcal{G},\mathcal{S}}^{(t)}$ denotes the marginal distribution of $\mu_{\mathcal{G}}^{(t)}$ on $\left\{\pi_j^{(t)} + \boldsymbol{n}_j^{(t)}\right\}_{j \in \mathcal{S}}$. We can see that for any $i \in \mathcal{S}$,

$$D_\alpha\left(\mu_{\mathcal{G},i}^{(T)}, \mu_{\mathcal{G}',i}^{(T)}\right) \le D_\alpha\left(\mu_{\mathcal{G},\mathcal{S}}^{(T)}, \mu_{\mathcal{G}',\mathcal{S}}^{(T)}\right).$$

For $i \notin \mathcal{S}$, we can show that $D_\alpha\left(\mu_{\mathcal{G},i}^{(T)}, \mu_{\mathcal{G}',i}^{(T)}\right)$ is bounded by some constant. Therefore, since $|\mathcal{S}| \ge N - 2$, $\frac{1}{N}\sum_{i=1}^N D_\alpha\left(\mu_{\mathcal{G},i}^{(T)}, \mu_{\mathcal{G}',i}^{(T)}\right)$ is bounded when $D_\alpha\left(\mu_{\mathcal{G},\mathcal{S}}^{(T)}, \mu_{\mathcal{G}',\mathcal{S}}^{(T)}\right)$ is sublinear in $N$.

Note that $\mu_{\mathcal{G},\mathcal{S}}^{(t)}(\cdot \mid \widehat{o}^{(1)}, \cdots, \widehat{o}^{(t-1)})$ is a multivariate Gaussian distribution with mean $\left(\pi_i^{(t)}\right)_{i \in \mathcal{S}}$ and variance $\sigma^2 \boldsymbol{I}^{\times_{i \in \mathcal{S}} \mathcal{A}_i}$. Therefore, by the Rényi divergence of multivariate Gaussian (Gil et al., 2013), the divergence is bounded by $\frac{\alpha}{2\sigma^2}\sum_{i \in \mathcal{S}}\left\|\pi_i^{(t)} - {\pi'}_i^{(t)}\right\|^2$.

If $\mathcal{S} = [N]$, then for any player $i \in [N] \setminus \{v_1, v_2\}$, given all past observations $\left(\left\{\pi_j^{(s)} + \boldsymbol{n}_j^{(s)}\right\}_{j \in \mathcal{S}}\right)_{s=1}^{t-1}$ are identical in $\mathcal{G}, \mathcal{G}'$, $\pi_i^{(t)} = {\pi'}_i^{(t)}$. For $i \in \{v_1, v_2\}$, given $\overline{\pi}_i^{(t-1)} = \overline{\pi}'_i^{(t-1)}$ and only $\boldsymbol{U}_{v_1,v_2} \ne \boldsymbol{U}'_{v_1,v_2}$, $\left\|\pi_i^{(t)} - {\pi'}_i^{(t)}\right\| \le \mathcal{O}\left(\frac{1}{|\mathcal{N}(i)|}\right) \le \mathcal{O}\left(\frac{1}{(\overline{\mathcal{N}})^{2/9}}\right)$ by definition of $\mathcal{S}$. $\mathcal{O}$ hides constants invariant to the number of players $N$, *e.g.* $A$, the size of the largest action set.

If $v_1, v_2 \notin \mathcal{S}$, then for any $i \in \mathcal{S} \setminus (\mathcal{N}(v_1) \cup \mathcal{N}(v_2))$, $\pi_i^{(t)} = {\pi'}_i^{(t)}$. For $i \in \mathcal{N}(v_1) \cup \mathcal{N}(v_2)$, we will further augment the space of observations so that the adversary can observe $\left(\left\{\boldsymbol{n}_{v_1}^{(s)}, \boldsymbol{n}_{v_2}^{(s)}\right\}\right)_{s=1}^{t-1}$. Assume $i \in \mathcal{N}(v_1) \setminus \mathcal{N}(v_2)$ for ease of exposition. For any $j \in \mathcal{N}(i) \setminus \{v_1, v_2\}$, we have $\overline{\pi}_j^{(t-1)} = \overline{\pi}'_j^{(t-1)}$. Therefore, $\left\|\pi_i^{(t)} - {\pi'}_i^{(t)}\right\| \le \mathcal{O}\left(\left\|\overline{\pi}_{v_1}^{(t-1)} - \overline{\pi}'_{v_1}^{(t-1)}\right\|\right) \le \mathcal{O}\left(\left\|\pi_{v_1}^{(t-1)} - {\pi'}_{v_1}^{(t-1)}\right\|\right)$. Moreover, due to the additional regularization imposed on each player, we have the following lemma.

**Lemma D.2.** *Consider the Algorithm 1. For any player $i \in [N]$ and timestep $t > 0$, by updating the strategy $\left\{\pi_i^{(s)}\right\}_{s=0}^t, \left\{{\pi'}_i^{(s)}\right\}_{s=0}^t$ with the same noise $\left\{\boldsymbol{n}_i^{(s)}\right\}_{s=0}^t$ but different gradients $\left\{\boldsymbol{g}_i^{(s)}\right\}_{s=0}^t, \left\{{\boldsymbol{g}'}_i^{(s)}\right\}_{s=0}^t$ individually, we have*

$$\left\|\pi_i^{(t)} - {\pi'}_i^{(t)}\right\| \le \frac{2\sqrt{\mathcal{A}_i}}{\tau_i}. \tag{28}$$

The proof is postponed to Appendix D.5. Therefore, finally, $\left\|\pi_i^{(t)} - {\pi'}_i^{(t)}\right\| \le \mathcal{O}\left(\frac{1}{\tau_{v_1}}\right) \le \mathcal{O}\left(\frac{\log N}{(\overline{\mathcal{N}})^{1/3}}\right)$ because $|\mathcal{N}(v_1)| \le (\overline{\mathcal{N}})^{2/9}$. Moreover, since $|\mathcal{N}(v_1)| \le (\overline{\mathcal{N}})^{2/9}$,

$$\sum_{i \in \mathcal{S}}\left\|\pi_i^{(t)} - {\pi'}_i^{(t)}\right\|^2 = \sum_{i \in \mathcal{S} \cap (\mathcal{N}(v_1) \cup \mathcal{N}(v_2))}\left\|\pi_i^{(t)} - {\pi'}_i^{(t)}\right\|^2 \le \mathcal{O}\left(\frac{(\log N)^2}{(\overline{\mathcal{N}})^{4/9}}\right).$$

For cases of $v_1 \in \mathcal{S}, v_2 \notin \mathcal{S}$ and $v_1 \notin \mathcal{S}, v_2 \in \mathcal{S}$, the proof is similar.

## D.2 Proof Sketch: the Case of Sparse Graphs (♠)

For the sparse graph, the degree of all players is bounded by a constant $\mathcal{N}^{\max}$. For simplicity, let's consider two adjacent games $\mathcal{G} \sim \mathcal{G}'$ which differs at the utility matrices on edge $(v_1, v_2) \in E$, and $\sigma = 0$ in Algorithm 1, *i.e.* the noise-free case.

At timestep $t$, $\pi_i^{(t)}$ in $\mathcal{G}$ and its counterpart $\pi_i'^{(t)}$ in $\mathcal{G}'$ are identical if $t \leq \min\{\text{dist}(i, v_1), \text{dist}(i, v_2)\}$. This can be proved by mathematical induction.

Therefore, Rényi divergence between $\mu_{\mathcal{G},i}^{(t)}(\cdot \mid \widehat{o}^{(1)}, \cdots, \widehat{o}^{(t-1)})$ and $\mu_{\mathcal{G}',i}^{(t)}(\cdot \mid \widehat{o}^{(1)}, \cdots, \widehat{o}^{(t-1)})$ is 0, since they are both normal distribution with mean $\pi_i^{(t)}$ and variance 0.

Recall that the graph is sparse and the degree of each player is bounded by a constant $\mathcal{N}^{\max}$. In such cases, most players are at least of distance $\mathcal{O}\left(\log_{\mathcal{N}^{\max}} N\right)$ to the edge $(v_1, v_2)$. Therefore, if $T$ in Algorithm 1 is no larger than $\mathcal{O}\left(\log_{\mathcal{N}^{\max}} N\right)$, the Rényi divergence of most players at timestep $t$ is 0. Therefore, by averaging across all players, the privacy budget will be 0 as the number of players goes to infinity.

The general proof for the cases when $\sigma > 0$ is shown in Appendix D.4.

In the following, we will introduce the proof for dense graphs (♣) first and that for the sparse graphs (♠) later.

## D.3 Proof for Dense Graphs (♣)

Let $\overline{\mathcal{N}} := \left(\frac{1}{N} \sum_{i \in [N]} \frac{1}{|\mathcal{N}(i)|}\right)^{-1}$ be the harmonic mean of players' degrees. When the graph is dense, $\overline{\mathcal{N}}$ will be relatively large. Now, we will bound Rényi DP. Let $(v_1, v_2)$ be the edge on which the utility matrix differs in $\mathcal{G}$ and $\mathcal{G}'$. Then, we define the set $\mathcal{S}$ as

$$
\mathcal{S} := \begin{cases}
[N] & |\mathcal{N}(v_1)|, |\mathcal{N}(v_2)| \geq \left(\overline{\mathcal{N}}\right)^{2/9} \\
[N] \setminus \{v_1\} & |\mathcal{N}(v_1)| < \left(\overline{\mathcal{N}}\right)^{2/9}, |\mathcal{N}(v_2)| \geq \left(\overline{\mathcal{N}}\right)^{2/9} \\
[N] \setminus \{v_2\} & |\mathcal{N}(v_2)| < \left(\overline{\mathcal{N}}\right)^{2/9}, |\mathcal{N}(v_1)| \geq \left(\overline{\mathcal{N}}\right)^{2/9} \\
[N] \setminus \{v_1, v_2\} & |\mathcal{N}(v_1)|, |\mathcal{N}(v_2)| < \left(\overline{\mathcal{N}}\right)^{2/9}.
\end{cases}
$$

Consider the marginal distribution $\mu_{\mathcal{G},\mathcal{S}}^{(t)}$ of $\mu_{\mathcal{G}}^{(t)}$ over $\left(\left\{\pi_i^{(s)} + \boldsymbol{n}_i^{(s)}\right\}_{i \in \mathcal{S}}\right)_{s=1}^{t}$. Then, $D_\alpha\left(\mu_{\mathcal{G},\mathcal{S}}^{(t)}, \mu_{\mathcal{G}',\mathcal{S}}^{(t)}\right) \geq D_\alpha\left(\mu_{\mathcal{G},i}^{(t)}, \mu_{\mathcal{G}',i}^{(t)}\right)$ for any $i \in [N] \setminus \{v_1, v_2\}$ and $t > 0$ since $i \in \mathcal{S}$ and enlarging the set of observations will not decrease the Rényi divergence. This is because post-processing (delete observations) will not increase Rényi divergence (Mironov, 2017).

### D.3.1 $v_1, v_2 \in \mathcal{S}$

By Lemma D.1, let $\pi_i'^{(t)}$ be the counterpart of $\pi_i^{(t)}$ for any $i \in [N]$, we have

$$
D_\alpha\left(\mu_{\mathcal{G},\mathcal{S}}^{(T)}, \mu_{\mathcal{G}',\mathcal{S}}^{(T)}\right) \leq \sum_{t=1}^{T} \sup_{\left(\widehat{o}^{(1)}, \widehat{o}^{(2)}, \cdots, \widehat{o}^{(t-1)}\right)} D_\alpha\left(\mu_{\mathcal{G},\mathcal{S}}(\cdot \mid \widehat{o}^{(1)}, \widehat{o}^{(2)}, \cdots, \widehat{o}^{(t-1)}), \mu_{\mathcal{G}',\mathcal{S}}(\cdot \mid \widehat{o}^{(1)}, \widehat{o}^{(2)}, \cdots, \widehat{o}^{(t-1)})\right)
$$

$$
\overset{(i)}{=} \frac{\alpha}{2\sigma^2} \sum_{t=1}^{T} \sum_{i \in [N]} \sup_{\left(\widehat{o}^{(1)}, \widehat{o}^{(2)}, \cdots, \widehat{o}^{(t-1)}\right)} \left\|\pi_i^{(t)} - \pi_i'^{(t)}\right\|^2.
$$

$(i)$ is by the Rényi divergence of multi-variate Gaussian distribution (Gil et al., 2013), since the distribution of $\pi_i^{(t)} + \boldsymbol{n}_i^{(t)}$ is a multivariate gaussian with mean $\pi_i^{(t)}$ and variance $\sigma^2 I_{\mathcal{A}_i}$ ($I_{\mathcal{A}_i}$ is the identity matrix indexed by $\mathcal{A}_i \times \mathcal{A}_i$). The inequality above implies that the Rényi divergence can be bounded by the squared 2-norm of $\pi_i^{(t)} - \pi_i'^{(t)}$, given all past observations $\left(\widehat{o}^{(1)}, \widehat{o}^{(2)}, \cdots, \widehat{o}^{(t-1)}\right)$ are identical in $\mathcal{G}, \mathcal{G}'$.

Given the observations are identical in $\mathcal{G}, \mathcal{G}'$, the gradient $\bar{\boldsymbol{g}}_i^{(t-1)}$ of all players except $v_1, v_2$ are identical in $\mathcal{G}, \mathcal{G}'$ since the gradient only depends on $\pi_i^{(t-1)} + \boldsymbol{n}_i^{(t-1)}$ and the utility matrices, which are identical for $i \in [N] \setminus \{v_1, v_2\}$. Therefore, $\pi_i^{(t)} = \pi'_i^{(t)}$ for any $i \in [N] \setminus \{v_1, v_2\}$. For $i \in \{v_1, v_2\}$, we have

$$
\begin{aligned}
\left\| \pi_i^{(t)} - \pi'_i^{(t)} \right\| &= \left\| \mathrm{Proj}_{\Delta^{\mathcal{A}_i}} \left( \frac{\bar{\pi}_i^{(t-1)} - \eta \bar{\boldsymbol{g}}_i^{(t-1)}}{1 + \eta \tau_i} \right) - \mathrm{Proj}_{\Delta^{\mathcal{A}_i}} \left( \frac{\bar{\pi}_i^{(t-1)} - \eta \boldsymbol{g}'_i^{(t-1)}}{1 + \eta \tau_i} \right) \right\| \\
&\leq \left\| \frac{\bar{\pi}_i^{(t-1)} - \eta \bar{\boldsymbol{g}}_i^{(t-1)}}{1 + \eta \tau_i} - \frac{\bar{\pi}_i^{(t-1)} - \eta \boldsymbol{g}'_i^{(t-1)}}{1 + \eta \tau_i} \right\| \\
&\leq \eta \left\| \frac{1}{|\mathcal{N}(i)|} \sum_{j \in \mathcal{N}(i)} \left( \boldsymbol{U}_{i,j} - \boldsymbol{U}'_{i,j} \right) \bar{\pi}_j^{(t-1)} \right\| \\
&\overset{(i)}{\leq} \frac{2\eta}{|\mathcal{N}(i)|} \sqrt{|\mathcal{A}_i|}.
\end{aligned}
$$

$(i)$ is because $\boldsymbol{U}_{i,j} \in [-1,1]^{\mathcal{A}_i \times \mathcal{A}_j}$ and $\boldsymbol{U}_{i,j} \neq \boldsymbol{U}'_{i,j}$ only if $(i,j) = (v_1, v_2)$ or $(i,j) = (v_2, v_1)$. Therefore, since the distribution of $\pi_i^{(t)} + \boldsymbol{n}_i^{(t)}$ is actually a multivariate gaussian with mean $\pi_i^{(t)}$ and variance $\sigma^2 I_{\mathcal{A}_i}$, so that the Rényi divergence is bounded by $\left( \frac{2\alpha\eta^2}{\sigma^2 |\mathcal{N}(v_1)|^2} |\mathcal{A}_{v_1}| + \frac{2\alpha\eta^2}{\sigma^2 |\mathcal{N}(v_2)|^2} |\mathcal{A}_{v_2}| \right) T$. By the definition of $\mathcal{S}$, it is further bounded by $\frac{4\alpha\eta^2}{\sigma^2 (\overline{N})^{4/9}} AT$.

### D.3.2  $v_1, v_2 \notin \mathcal{S}$

When the adversary may only observe $\left( \pi_k^{(s)} + \boldsymbol{n}_k^{(s)} \right)_{s=1}^{t}$ for some $k \in \{v_1, v_2\}$, by Lemma D.1, we only need to bound $\left\| \pi_k^{(t)} - \pi'_k^{(t)} \right\|^2$ when all past observations $\left( \pi_k^{(s)} + \boldsymbol{n}_k^{(s)} \right)_{s=1}^{t-1}$ are the same in $\mathcal{G}, \mathcal{G}'$. Then,

$$
\begin{aligned}
\left\| \pi_k^{(t)} - \pi'_k^{(t)} \right\| &= \left\| \mathrm{Proj}_{\Delta^{\mathcal{A}_k}} \left( \frac{\bar{\pi}_k^{(t-1)} - \eta \boldsymbol{g}_k^{(t-1)}}{1 + \eta \tau_k} \right) - \mathrm{Proj}_{\Delta^{\mathcal{A}_k}} \left( \frac{\bar{\pi}_k^{(t-1)} - \eta \boldsymbol{g}'_k^{(t-1)}}{1 + \eta \tau_k} \right) \right\| \\
&\leq \left\| \frac{\bar{\pi}_k^{(t-1)} - \eta \boldsymbol{g}_k^{(t-1)}}{1 + \eta \tau_k} - \frac{\bar{\pi}_k^{(t-1)} - \eta \boldsymbol{g}'_k^{(t-1)}}{1 + \eta \tau_k} \right\| \\
&\leq \eta \left\| \boldsymbol{g}_k^{(t-1)} - \boldsymbol{g}'_k^{(t-1)} \right\| \\
&\leq 2\eta \sqrt{|\mathcal{A}_k|}.
\end{aligned}
$$

For $k \in [N] \setminus \{v_1, v_2\}$, similar to the proof in the previous section, we will augment the observations of the adversary first. The adversary may observe $\left( \left\{ \pi_i^{(s)} + \boldsymbol{n}_i^{(s)} \right\}_{i \in [N] \setminus \{v_1, v_2\}} \right)_{s=1}^{t}$ and $\left( \left\{ \boldsymbol{n}_{v_1}^{(s)}, \boldsymbol{n}_{v_2}^{(s)} \right\} \right)_{s=1}^{t}$ at timestep $t$. Note that the actual observation of the adversary is still $\left( \pi_k^{(s)} + \boldsymbol{n}_k^{(s)} \right)_{s=1}^{t}$. We augment the observations to simplify the proof, since the Rényi divergence of the the augmented observation's distribution upperbounds that of the actual observation.

Still, by the chain rule of Rényi divergence, we need to bound $\sum_{i \in [N] \setminus \{v_1, v_2\}} \left\| \pi_i^{(t)} - \pi'_i^{(t)} \right\|$ for any $t \in [T]$. For any $i \notin \mathcal{N}(v_1) \cup \mathcal{N}(v_2)$, similar to the discussion in previous section, the gradients $\bar{\boldsymbol{g}}_i^{(t)}$ and $\bar{\pi}_i^{(t-1)}$ are identical in $\mathcal{G}, \mathcal{G}'$, so that the $\pi_i^{(t)} = \pi'_i^{(t)}$.

For $i \in \mathcal{N}(v_1) \cup \mathcal{N}(v_2) \setminus \{v_1, v_2\}$, we have

$$
\begin{aligned}
\left\| \pi_i^{(t)} - \pi'_i^{(t)} \right\| &\leq \eta \left\| \bar{\boldsymbol{g}}_i^{(t-1)} - \boldsymbol{g}'_i^{(t-1)} \right\| = \frac{\eta}{|\mathcal{N}(i)|} \left\| \sum_{j \in \mathcal{N}(i)} \boldsymbol{U}_{i,j} \left( \bar{\pi}_j^{(t-1)} - \bar{\pi}'_j^{(t-1)} \right) \right\| \\
&\overset{(i)}{=} \frac{\eta}{|\mathcal{N}(i)|} \left\| \sum_{j \in \{v_1, v_2\} \cap \mathcal{N}(i)} \boldsymbol{U}_{i,j} \left( \bar{\pi}_j^{(t-1)} - \bar{\pi}'_j^{(t-1)} \right) \right\| \\
&\leq \frac{\eta \sqrt{|\mathcal{A}_i|}}{|\mathcal{N}(i)|} \sum_{j \in \{v_1, v_2\} \cap \mathcal{N}(i)} \left\| \bar{\pi}_j^{(t-1)} - \bar{\pi}'_j^{(t-1)} \right\|_1 \\
&\leq \frac{\eta A}{|\mathcal{N}(i)|} \sum_{j \in \{v_1, v_2\} \cap \mathcal{N}(i)} \left\| \bar{\pi}_j^{(t-1)} - \bar{\pi}'_j^{(t-1)} \right\|.
\end{aligned}
$$

$(i)$ uses the fact that $\bar{\pi}_i^{(t-1)} = \bar{\pi}'_i^{(t-1)}$ for $i \notin \{v_1, v_2\}$. Since, $\boldsymbol{n}_j^{(t-1)} = \boldsymbol{n}'_j^{(t-1)}$, for any $j \in \{v_1, v_2\}$, we have

$$
\begin{aligned}
\left\| \bar{\pi}_j^{(t-1)} - \bar{\pi}'_j^{(t-1)} \right\| &= \left\| \mathrm{Proj}_{\Delta^{\mathcal{A}_j}} \left( \pi_j^{(t-1)} + \boldsymbol{n}_j^{(t-1)} \right) - \mathrm{Proj}_{\Delta^{\mathcal{A}_j}} \left( \pi'_j^{(t-1)} + \boldsymbol{n}'_j^{(t-1)} \right) \right\| \\
&\leq \left\| \pi_j^{(t-1)} + \boldsymbol{n}_j^{(t-1)} - \pi'_j^{(t-1)} - \boldsymbol{n}'_j^{(t-1)} \right\| \\
&= \left\| \pi_j^{(t-1)} - \pi'_j^{(t-1)} \right\|.
\end{aligned}
$$

Moreover, due to the additional regularization, for $j \in \{v_1, v_2\}$, $\left\| \pi_j^{(t-1)} - \pi'_j^{(t-1)} \right\|$ will be bounded as follows according to Lemma D.2.

$$
\left\| \pi_j^{(t-1)} - \pi'_j^{(t-1)} \right\| \leq \frac{2\sqrt{\mathcal{A}_j}}{\tau_j} \overset{(i)}{\leq} 2\sqrt{\mathcal{A}_j} \frac{\left( \overline{\mathcal{N}} \right)^{2/9} \log N}{\left( \overline{\mathcal{N}} \right)^{5/9}} = \frac{2\sqrt{\mathcal{A}_j} \log N}{\left( \overline{\mathcal{N}} \right)^{1/3}}.
$$

$(i)$ is because when $v_1 \notin \mathcal{S}$, $|\mathcal{N}(v_1)| < \left( \overline{\mathcal{N}} \right)^{2/9}$ so that $\tau_{v_1} \geq \frac{\left( \overline{\mathcal{N}} \right)^{5/9}}{\left( \overline{\mathcal{N}} \right)^{2/9} \log N}$ by definition. The same argument also holds for $v_2$. Therefore, $\left\| \pi_i^{(t)} - \pi'_i^{(t)} \right\| \leq \frac{4\eta A^{3/2} \log N}{\left( \overline{\mathcal{N}} \right)^{1/3} |\mathcal{N}(i)|}$, where $A = \max_{i \in [N]} |\mathcal{A}_i|$. Furthermore, for $i \in [N] \setminus \{v_1, v_2\}$

$$
\begin{aligned}
D_\alpha \left( \mu_{\mathcal{G},i}^{(T)}, \mu_{\mathcal{G}',i}^{(T)} \right) \leq D_\alpha \left( \mu_{\mathcal{G},\mathcal{S}}^{(T)}, \mu_{\mathcal{G}',\mathcal{S}}^{(T)} \right) &\leq \frac{\alpha}{2\sigma^2} |\mathcal{N}(v_1) \cup \mathcal{N}(v_2)| \cdot \left( \frac{16\eta^2 A^3 (\log N)^2}{\left( \overline{\mathcal{N}} \right)^{2/3} \min_{i \in \mathcal{N}(v_1) \cup \mathcal{N}(v_2)} |\mathcal{N}(i)|^2} \right) T \\
&\leq \frac{\alpha}{2\sigma^2} \left( 2 \left( \overline{\mathcal{N}} \right)^{2/9} \right) \cdot \left( \frac{16\eta^2 A^3 (\log N)^2}{\left( \overline{\mathcal{N}} \right)^{2/3} \min_{i \in \mathcal{N}(v_1) \cup \mathcal{N}(v_2)} |\mathcal{N}(i)|^2} \right) T \\
&= \frac{16\alpha\eta^2 A^3 (\log N)^2}{\sigma^2 \left( \overline{\mathcal{N}} \right)^{4/9} \min_{i \in \mathcal{N}(v_1) \cup \mathcal{N}(v_2)} |\mathcal{N}(i)|^2} T.
\end{aligned}
$$

Finally, since $\min_{i \in \mathcal{N}(v_1) \cup \mathcal{N}(v_2)} |\mathcal{N}(i)|^2 \geq 1$, we can conclude the proof. Moreover, $\min_{i \in \mathcal{N}(v_1) \cup \mathcal{N}(v_2)} |\mathcal{N}(i)|^2$ can be much larger than a constant in practice, *e.g.,* Erdős–Rényi graphs.

The proofs for the rest possibilities ($v_1 \in \mathcal{S}, v_2 \notin \mathcal{S}$ and $v_1 \notin \mathcal{S}, v_2 \in \mathcal{S}$) are similar. $\qquad \square$

### D.4 PROOF FOR SPARSE GRAPHS (♠)

In this section, we show the DP guarantee when the associated graph of the polymatrix game is sparse. Formally, there exists a constant $\mathcal{N}^{\max} > 0$ so that $|\mathcal{N}(i)| \leq \mathcal{N}^{\max}$ for any $i \in [N]$.

We will augment the observations from $\left( \pi_i^{(s)} + \boldsymbol{n}_i^{(s)} \right)_{s=1}^{t-1}$ to $\left( \left\{ \pi_j^{(s)} + \boldsymbol{n}_j^{(s)} \right\}_{j \in \mathcal{S}_i^{(s)}} \right)_{s=1}^{t}$, where

$$
\mathcal{S}_i^{(s)} := \begin{cases} \{j: \ \min \{\mathrm{dist}(j, v_1), \mathrm{dist}(j, v_2)\} \geq s\} & \min \{\mathrm{dist}(i, v_1), \mathrm{dist}(i, v_2)\} \geq s \\ \{i\} & \text{Otherwise.} \end{cases}
$$

When $s \leq \min\{\text{dist}(i, v_1), \text{dist}(i, v_2)\}$, we will show in the following that with all past observations identical in $\mathcal{G}, \mathcal{G}'$, then $\pi_j^{(s)}$ and its counterpart $\pi'_j^{(s)}$ in $\mathcal{G}'$ are identical for any $j \in \mathcal{S}_i^{(s)}$. When $s = 0$, $\mathcal{S}_i^{(0)} = [N]$ and $\pi_j^{(0)} = \pi'_j^{(0)}$ for any $j \in [N]$ since they are both initialized to uniform distribution over $\mathcal{A}_j$. When $s > 0$, for any $j \in \mathcal{S}_i^{(s)}$, we must have $\mathcal{N}(j) \subseteq \mathcal{S}_i^{(s-1)}$, otherwise $\min\{\text{dist}(j, v_1), \text{dist}(j, v_2)\} < s$. Therefore, since all past observations are identical, we have $\bar{\pi}_k^{(s-1)}$ are identical in $\mathcal{G}, \mathcal{G}'$ for any $k \in \mathcal{N}(j)$ so that the gradient $g_j^{(s-1)} = g'_j^{(s-1)}$. Moreover, given $\bar{\pi}_j^{(s-1)}$ are identical, $\pi_j^{(s)}$ are identical in $\mathcal{G}, \mathcal{G}'$.

When $s > \min\{\text{dist}(i, v_1), \text{dist}(i, v_2)\}$, $\left\| \pi_i^{(s)} - \pi'_i^{(s)} \right\| \leq 2\eta\sqrt{|\mathcal{A}_i|}$, given the past observation $\pi_i^{(s-1)} + n_i^{(s-1)}$ is identical.

By Lemma D.1, for any $T > 0$,

$$D_\alpha\left(\mu_{\mathcal{G},i}^{(T)}, \mu_{\mathcal{G}',i}^{(T)}\right) \leq \max\{0, T - \min\{\text{dist}(i, v_1), \text{dist}(i, v_2)\}\} \frac{2\alpha\eta^2}{\sigma^2}|\mathcal{A}_i|$$

$$\leq \mathbb{1}\left(T > \min\{\text{dist}(i, v_1), \text{dist}(i, v_2)\}\right) \frac{2\alpha\eta^2}{\sigma^2}|\mathcal{A}_i|T. \qquad \square$$

### D.5 OMITTED PROOF OF LEMMAS

**Lemma D.2.** *Consider the Algorithm 1. For any player $i \in [N]$ and timestep $t > 0$, by updating the strategy $\left\{\pi_i^{(s)}\right\}_{s=0}^t, \left\{\pi'_i^{(s)}\right\}_{s=0}^t$ with the same noise $\left\{n_i^{(s)}\right\}_{s=0}^t$ but different gradients $\left\{g_i^{(s)}\right\}_{s=0}^t, \left\{g'_i^{(s)}\right\}_{s=0}^t$ individually, we have*

$$\left\| \pi_i^{(t)} - \pi'_i^{(t)} \right\| \leq \frac{2\sqrt{\mathcal{A}_i}}{\tau_i}. \qquad (28)$$

*Proof.*

$$\left\| \pi_i^{(t+1)} - \pi'_i^{(t+1)} \right\| = \left\| \text{Proj}_{\Delta^{\mathcal{A}_i}}\left(\frac{\text{Proj}_{\Delta^{\mathcal{A}_i}}\left(\pi_i^{(t)} + n_i^{(t)}\right) - \eta\bar{g}_i^{(t)}}{1 + \eta\tau_i}\right) - \text{Proj}_{\Delta^{\mathcal{A}_i}}\left(\frac{\text{Proj}_{\Delta^{\mathcal{A}_i}}\left(\pi'_i^{(t)} + n_i^{(t)}\right) - \eta g'_i^{(t)}}{1 + \eta\tau_i}\right) \right\|$$

$$\leq \left\| \frac{\text{Proj}_{\Delta^{\mathcal{A}_i}}\left(\pi_i^{(t)} + n_i^{(t)}\right) - \eta\bar{g}_i^{(t)}}{1 + \eta\tau_i} - \frac{\text{Proj}_{\Delta^{\mathcal{A}_i}}\left(\pi'_i^{(t)} + n_i^{(t)}\right) - \eta g'_i^{(t)}}{1 + \eta\tau_i} \right\|$$

$$\leq \frac{1}{1 + \eta\tau_i}\left\| \text{Proj}_{\Delta^{\mathcal{A}_i}}\left(\pi_i^{(t)} + n_i^{(t)}\right) - \text{Proj}_{\Delta^{\mathcal{A}_i}}\left(\pi'_i^{(t)} + n_i^{(t)}\right) \right\| + \frac{\eta}{1 + \eta\tau_i}\left\| \bar{g}_i^{(t)} - g'_i^{(t)} \right\|$$

$$\leq \frac{1}{1 + \eta\tau_i}\left\| \pi_i^{(t)} - \pi'_i^{(t)} \right\| + \frac{\eta}{1 + \eta\tau_i}\left\| \bar{g}_i^{(t)} - g'_i^{(t)} \right\|.$$

By recursively applying the inequality above, we have

$$\left\| \pi_i^{(t+1)} - \pi'_i^{(t+1)} \right\| \leq \left(\frac{1}{1 + \eta\tau_i}\right)^{t+1}\left\| \pi_i^{(0)} - \pi'_i^{(0)} \right\| + \eta\sum_{s=0}^t\left(\frac{1}{1 + \eta\tau_i}\right)^{t-s+1}\left\| g_i^{(s)} - g'_i^{(s)} \right\|$$

$$\overset{(i)}{=} \eta\sum_{s=0}^t\left(\frac{1}{1 + \eta\tau_i}\right)^{t-s+1}\left\| g_i^{(s)} - g'_i^{(s)} \right\|.$$

$(i)$ is because $\pi_i^{(0)}, \pi'_i^{(0)}$ are both initialized as uniform distribution over $\mathcal{A}_i$. Therefore, since each element of the gradient is bounded by $[-1, 1]$ by definition, we have

$$\left\| \pi_i^{(t+1)} - \pi'_i^{(t+1)} \right\| \leq 2\eta\sum_{s=0}^t\left(\frac{1}{1 + \eta\tau_i}\right)^{t-s+1}\sqrt{\mathcal{A}_i} \leq 2\eta\sqrt{\mathcal{A}_i}\frac{1}{(1 + \eta\tau_i)\left(1 - \frac{1}{1+\eta\tau_i}\right)} = \frac{2\sqrt{\mathcal{A}_i}}{\tau_i}.$$

$\square$

**Lemma D.1** (Chain Rule of Rényi divergence). *For any distribution $p, q$ over random variables $X^1, X^2$, for any $\alpha \geq 1$, we have*

$$
\begin{aligned}
&D_\alpha \left( p(X^1, X^2), q(X^1, X^2) \right) \\
&\leq D_\alpha \left( p(X^1), q(X^1) \right) + \sup_{\widehat{x}^1} D_\alpha \left( p(X^2 \mid X^1 = \widehat{x}^1), q(X^2 \mid X^1 = \widehat{x}^1) \right).
\end{aligned} \tag{27}
$$

*Proof.* By definition, when $\alpha > 1$, we have

$$
\begin{aligned}
&D_\alpha \left( p(X^1, X^2), q(X^1, X^2) \right) \\
&= \frac{1}{\alpha - 1} \log \int_{x^1, x^2} \left( p(x^1, x^2) \right)^\alpha \left( q(x^1, x^2) \right)^{1-\alpha} dx^1 dx^2 \\
&\overset{(i)}{=} \frac{1}{\alpha - 1} \log \int_{x^1} \left( \int_{x^2} \left( p(x^2 \mid x^1) \right)^\alpha \left( q(x^2 \mid x^1) \right)^{1-\alpha} dx^2 \right) \left( p(x^1) \right)^\alpha \left( q(x^1) \right)^{1-\alpha} dx^1 \\
&\overset{(ii)}{\leq} \frac{1}{\alpha - 1} \log \sup_{\widehat{x}^1} \left( \int_{x^2} \left( p(x^2 \mid \widehat{x}^1) \right)^\alpha \left( q(x^2 \mid \widehat{x}^1) \right)^{1-\alpha} dx^2 \right) \left( \int_{x^1} \left( p(x^1) \right)^\alpha \left( q(x^1) \right)^{1-\alpha} dx^1 \right) \\
&= \frac{1}{\alpha - 1} \sup_{\widehat{x}^1} \log \int_{x^2} \left( p(x^2 \mid \widehat{x}^1) \right)^\alpha \left( q(x^2 \mid \widehat{x}^1) \right)^{1-\alpha} dx^2 + \frac{1}{\alpha - 1} \log \int_{x^1} \left( p(x^1) \right)^\alpha \left( q(x^1) \right)^{1-\alpha} dx^1 \\
&= D_\alpha \left( p(X^1), q(X^1) \right) + \sup_{\widehat{x}^1} D_\alpha \left( p(X^2 \mid X^1 = \widehat{x}^1), q(X^2 \mid X^1 = \widehat{x}^1) \right).
\end{aligned}
$$

$(i)$ is by Tonelli's theorem. $(ii)$ is by Hölder's Inequality.

When $\alpha = 1$, the corresponding Rényi divergence is KL-divergence. Therefore,

$$
\begin{aligned}
&D_1 \left( p(X^1, X^2), q(X^1, X^2) \right) \\
&= \int_{x^1, x^2} p(x^1, x^2) \log \frac{p(x^1, x^2)}{q(x^1, x^2)} dx^1 dx^2 \\
&= \int_{x^1, x^2} p(x^2 \mid x^1) p(x^1) \log \frac{p(x^2 \mid x^1) p(x^1)}{q(x^2 \mid x^1) q(x^1)} dx^1 dx^2 \\
&= \int_{x^1, x^2} p(x^2 \mid x^1) p(x^1) \log \frac{p(x^1)}{q(x^1)} dx^1 dx^2 + \int_{x^1, x^2} p(x^2 \mid x^1) p(x^1) \log \frac{p(x^2 \mid x^1)}{q(x^2 \mid x^1)} dx^1 dx^2 \\
&= D_1 \left( p(X^1), q(X^1) \right) + \mathbb{E}_{\widehat{x}^1 \sim p(X^1)} \left[ D_1 \left( p(X^2 \mid X^1 = \widehat{x}^1), q(X^2 \mid X^1 = \widehat{x}^1) \right) \right] \\
&\leq D_1 \left( p(X^1), q(X^1) \right) + \sup_{\widehat{x}^1} D_1 \left( p(X^2 \mid X^1 = \widehat{x}^1), q(X^2 \mid X^1 = \widehat{x}^1) \right).
\end{aligned}
$$

$\square$

# E EXPERIMENTS

The experimental results are shown in Figures 5 and 6. The baseline algorithm is adapted from Huang et al. (2015). Our implementation uses PyTorch (Paszke et al., 2019) to enable efficient, fully parallel updates of all players' strategies, and all runs were executed on $2\times$ NVIDIA H200 GPUs. The error bars denote $1\sigma$.

On dense graphs, as the number of players increases, our algorithm's exploitability and the privacy budget both decrease, whereas the baseline's exploitability increases. On sparse graphs, by contrast, only the privacy budget decreases. A plausible explanation is that the convergence rate on sparse graphs is $\mathcal{O}\left( \frac{1}{(\log N)^{1/3}} \right)$, which is too mild to overcome constant factors and stochastic noise for $N \leq 2^{16}$. Hyperparameters were chosen according to Theorem 7.1.

## E.1 RESULTS IN GRAPHS WITH DIFFERENT TOPOLOGIES

In this section, we provide further results in dense and sparse graphs with different topologies.

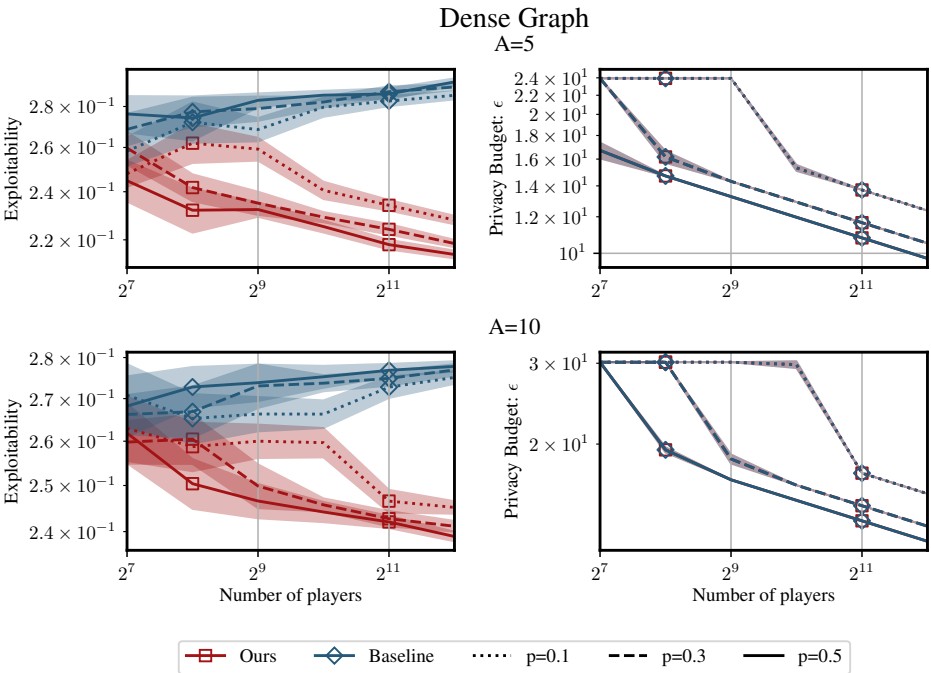

Figure 5: Experiment results of dense graphs. Each node (player) in the graph will be connected to another node with probability $p$, the connection is sampled i.i.d. for each node. Then, the duplicate edges will be removed. The action set sizes of all players are set to $A$. We can see from the result that the exploitability and the privacy budget both decrease as the number of players increases.

- Dense graph: players are randomly divided into $\lfloor 1/p \rfloor$ clusters. All pairs of players within the same cluster interact with one another, while pairs of players belonging to different clusters interact with probability $\frac{10p}{N}$.
- Sparse graph: players are randomly sampled on a 2-D plane. Each player only interacts with the $c$-nearest neighbors.

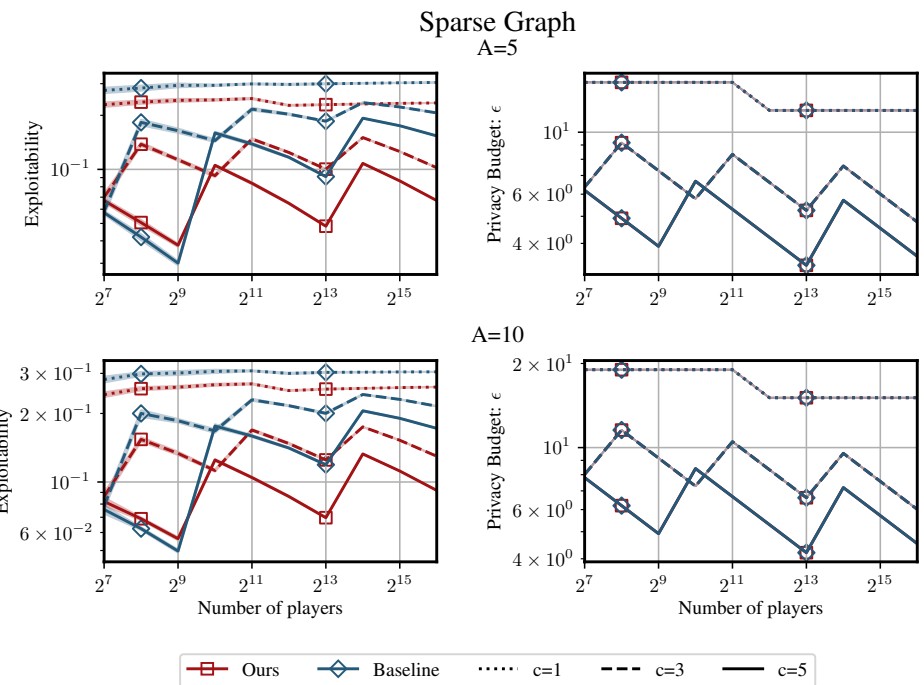

Figure 6: Experiment results of sparse graphs. We will randomly generate $cN$ edges in total, with each node appearing $c$ times. Then, duplicate edges and self-loops will be removed. The action set sizes of all players are set to $A$. The result shows that while the exploitability remains unchanged, the exploitability decreases as the number of players increases.

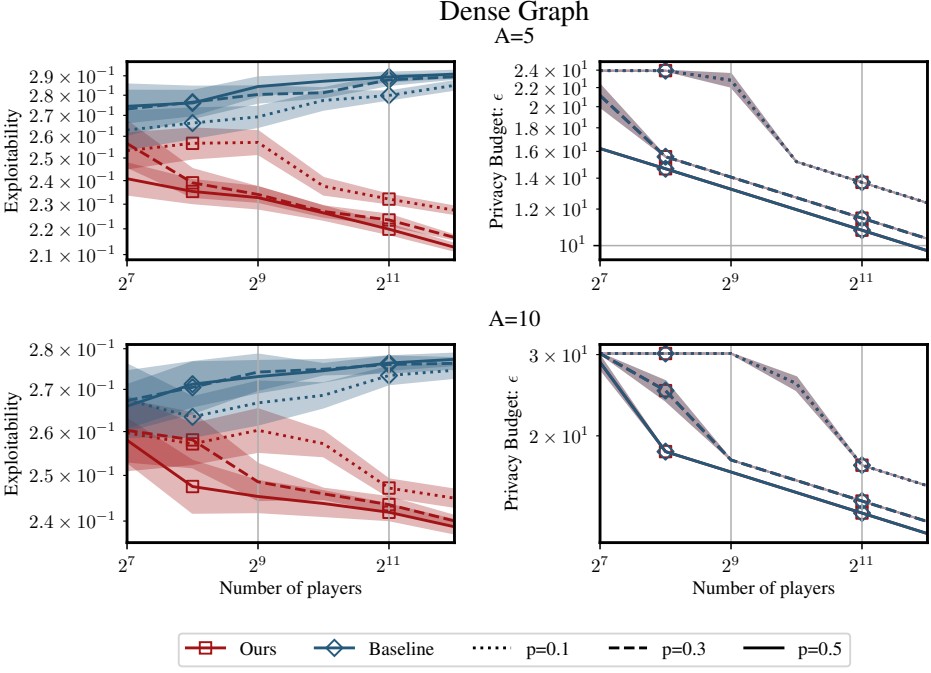

Figure 7: Players are randomly divided into $\lfloor 1/p \rfloor$ clusters. All pairs of players within the same cluster interact with one another, while pairs of players belonging to different clusters interact with probability $\frac{10p}{N}$. The action set of each player is of size $A$. We can see from the result that the exploitability and the privacy budget both decrease as the number of players increases.

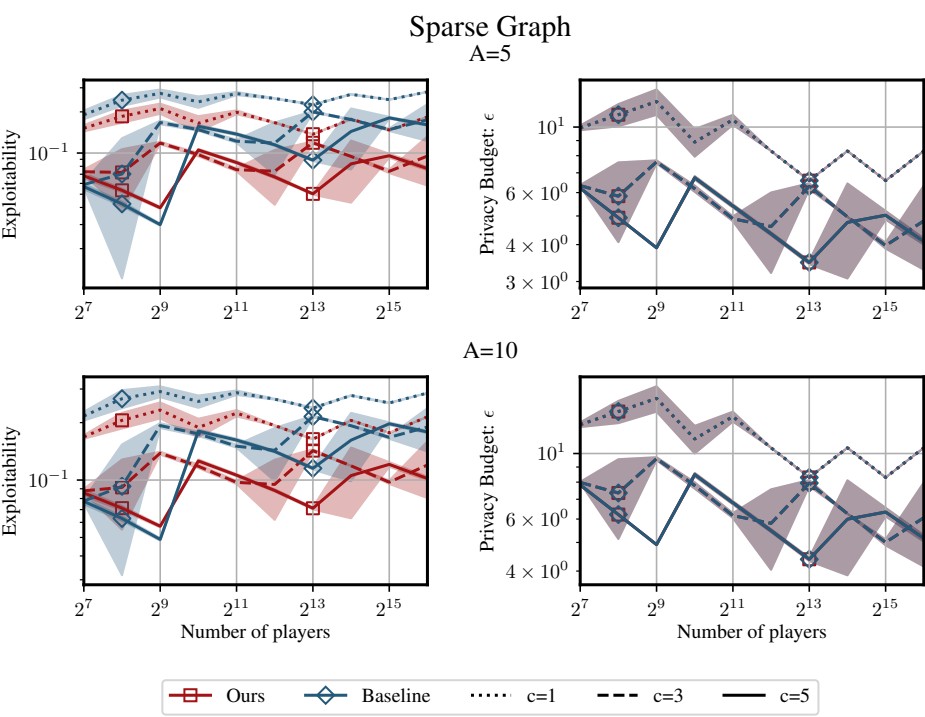

Figure 8: Players are randomly sampled on a 2-D plane. Each player only interacts with the $c$-nearest neighbors. The action set of each player is of size $A$. The result shows that while the exploitability remains unchanged, the exploitability decreases as the number of players increases.

