# OpenReview forum: "Differentially Private Equilibrium Finding in Polymatrix Games"
_ICLR.cc/2026/Conference — ICLR 2026 Poster_

### Official Review · Reviewer_eMY6 · 2025-10-31

**Soundness:** 2
**Presentation:** 3
**Contribution:** 2
**Rating:** 4
**Confidence:** 4

**Summary:**

This work proves that for polymatrix games, if either adversary can access an arbitrary number of communication channels or the learning objective is to minimize the Euclidean distance to the equilibrium set, no algorithm can simultaneously find an accurate enough approximate equilibrium strategy profile and guarantee a low level of privacy. The authors further relax the restrictions to consider that the adversary can only access a bounded number of channels. Under this condition, the authors propose an algorithm that can simultaneously find an approximate coarse correlated equilibrium (CCE) strategy profile and guarantee a low level of privacy. Experiments with a varying number of players are conducted to validate the effectiveness of the proposed algorithm.

**Strengths:**

The presentation of most parts of the paper is clear.

**Weaknesses:**

I’m concerned with the problem formulation, the motivation, and the design of the proposed algorithm.

First, it is unclear to me whether the utility matrix $U\_{i,j}$ for each $ (i, j) \in E $ is known or unknown in Section 2. I guess the authors assume known $U\_{i,j}$’s as the proposed algorithm explicitly uses $U\_{i,j}$’s to compute the gradient $\overline{\boldsymbol{g}}\_i^{(t)}$. However, if each $U\_{i,j}$ is known, I think this problem is just an optimization problem and the equilibrium can be **computed** simply using an optimization method in an offline manner. In this sense, there is no need to use online learning to **learn** an approximate equilibrium strategy profile in an online manner. Consequently, I think no communications between agents and privacy-preserving are needed.

Further, even if each $U\_{i,j}$ is known, there are many works in the literature studying how to learn the approximate equilibrium strategy profile under the bandit feedback (i.e., only the rewards of the chosen actions are revealed) in an uncoupled manner (say, [1,2]). These algorithms do not require any communication or coordination between the agents. As such, there is also no need to preserve privacy in these scenarios.

Therefore, I think the motivation for studying privacy-preserving algorithms using the communication of the policy information between agents seems not very reasonable. This also casts doubt on the value of the proposed algorithm.

[1] Cai et al. Uncoupled and Convergent Learning in Two-Player Zero-Sum Markov Games with Bandit Feedback. NeurIPS, 23.

[2] Ito et al. Instance-Dependent Regret Bounds for Learning Two-Player Zero-Sum Games with Bandit Feedback. COLT, 25.

**Questions:**

Please see my questions in the weakness part.

---

> ### Author Response · Authors · 2025-11-20
> **Response to Reviewer eMY6**
>
> We thank the reviewer for their thoughtful comments. We appreciate the opportunity to clarify the problem formulation regarding the privacy of utility matrices and the necessity of privacy-preserving mechanisms even in uncoupled settings.
>
> We believe most of the weaknesses stem from misunderstandings that are easy to rectify; we hope the discussion here resolves them and improves your opinion of the paper.
>
> ## Who knows $U_{i,j}$
>
> The utility matrix $U_{i,j}$ is known **only to player $i$**, who wishes to keep it private. Consequently, a centralized offline algorithm is infeasible. This setting is consistent with the literature on decentralized optimization with differential privacy guarantees (*e.g.*, [3]).
>
> ## Learning with Bandit Feedback Leaks Information
>
> The reviewer suggests that uncoupled bandit algorithms (*e.g.*, [1][2]) inherently preserve privacy because they do not require explicit communication. This is not the case: **lack of explicit communication does not imply privacy.**
>
> Indeed, consider the following counterexample:
>
> Consider a simple game where Player 1 has a singleton action set $\{x_1\}$ and Player 2 has two actions $\{a, b\}$.
>
> - Let Player 1’s private utility matrix be $[[0, 1]]$.
> - Let Player 2’s private utility matrix be $[[u_a], [u_b]]$.
> - In a standard uncoupled learning setting (such as [1] or [2]), Player 2 will eventually converge to the action with higher utility (*e.g.*, action $a$ if $u_a > u_b$).
>
> **From the reward received, Player 1 can infer Player 2’s action: a reward of 0 indicates that Player 2 chose action $a$, while a reward of 1 indicates that Player 2 chose action $b$.**
>
> By simply observing Player 2's trajectory, Player 1 (or an adversary) can infer that $u_a > u_b$. If the algorithm is deterministic or follows a known update rule (*e.g.*, Hedge or projected gradient ascent), an adversary observing the frequency of updates can even estimate the numerical values of $u_a$ and $u_b$ from the frequency of picking $a,b$.
>
> Hence, even bandit algorithms leak the private utility matrix $U$, since players' actions include the information about $U$. To prevent an adversary from reconstructing $U$ based on the observed history of play, we must introduce privacy-preserving mechanisms (noise). However, naively adding noise destroys the solution accuracy.
>
> Our paper explicitly addresses this trade-off: we utilize the structure of polymatrix games to design a mechanism that ensures differential privacy while maintaining convergence to an approximate equilibrium, a guarantee that standard uncoupled algorithms cannot provide.
>
> [1] Cai et al. Uncoupled and Convergent Learning in Two-Player Zero-Sum Markov Games with Bandit Feedback. NeurIPS, 23.
>
> [2] Ito et al. Instance-Dependent Regret Bounds for Learning Two-Player Zero-Sum Games with Bandit Feedback. COLT, 25.
>
> [3] Huang, Zhenqi, Sayan Mitra, and Nitin Vaidya. "Differentially private distributed optimization." Proceedings of the 16th International Conference on Distributed Computing and Networking. 2015.

---

> > ### Comment · Reviewer_eMY6 · 2025-11-28
> >
> > I’d like to thank the authors for their responses. However, I’m still not fully convinced by the justification, for the following reasons:
> >
> > 1. Even if the utility matrix $U_{i,j}$ is known only to player $i$, I think an alternative way to address the privacy concerns is to use a centralized trusted third party who can receive the utility matrices of all individuals but not reveal the information of player $i$ to any player $j \neq i$. I’d like to suggest that the authors motivate and explain, in the revision of the paper, the advantages of the decentralized method with differential privacy in this work over the centralized method above.
> > 2. Further, the authors said that “players’ actions include information about U”. I fully agree with this, whereas the references I provided in my initial review do not require the max-player to observe the min-player’s actions; that is, their algorithms are fully uncoupled. Therefore, even in the absence of the trusted third party mentioned in Point 1, I think using the algorithms in the references I pointed out earlier should still not suffer from the privacy concerns.

---

> ### Author Response · Authors · 2025-11-28
>
> We thank the reviewer for the prompt engagement and the opportunity to clarify these crucial points. We address the two remaining concerns below.
>
> ## Motivation: Decentralized vs. Centralized (Trusted Third Party)
>
> > I’d like to suggest that the authors motivate and explain, in the revision of the paper, the advantages of the decentralized method with differential privacy in this work over the centralized method above.
>
> In Lines 43–50 (line 40-49 of the initial submission) of the current version, we explain the motivation for adopting a decentralized approach other than a centralized one. In brief, we favor decentralization because a trusted central server is not always feasible.
>
> ## Technical Clarification: Uncoupled Bandit Algorithms Leak Privacy
>
> The reason that references [1] and [2] leak information is illustrated by the example provided in our initial rebuttal. We elaborate on it here.
>
> - Player 1 receives a reward at each timestep (either 0 or 1).
> - This reward depends on the **joint action** of all players: a reward of 0 indicates that Player 2 chose action $a$, while a reward of 1 indicates that Player 2 chose action $b$.
> - Consequently, from the reward alone, Player 1 can deduce the **joint action**, and thus directly infer Player 2’s individual action.
>
> Hence, Player 1 can infer Player 2’s action at each timestep and, consequently, deduce Player 2’s utility matrix.

---

### Official Review · Reviewer_UsT1 · 2025-10-31

**Soundness:** 3
**Presentation:** 3
**Contribution:** 3
**Rating:** 6
**Confidence:** 3

**Summary:**

This paper studies equilibrium computation in polymatrix games under edge-DP (everything else is fixed in the neighboring "games" execpt the value of a single edge). The authors observe that existing works cannot simultaneously achieve high accuracy meaning good approximation to equilibrium) and low privacy budget in fully distributed setting.

The paper gives impossibility results which show that achieving vanishing privacy loss and vanishing error simultaneously is impossible under two regimes: when the accuracy metric is Euclidean, or when the adversary can access all communication channels.

Under more restricted conditions where an adversary can access only a limited number of channels, and accuracy measured by a certain exploitability metric, they design a distributed, DP algorithm where each player sends a noisy version of their message to neighbors and updates using a regularized proximal gradient step with an adaptive regularizer scaled by the player’s degree.

It is then shown that for this algorithm, both the expected exploitability and the privacy budget vanish as the number of players $N$ grows, in both sparse and dense graphs.

**Strengths:**

-rigorous formalization of distributed equilibrium computation with DP using the adjacency notion from [Huang et al. (2015)](https://arxiv.org/pdf/1401.2596) (edge-DP).

- The impossibility theorems are conceptual but insightful: they clarify why prior work could not achieve both accuracy and privacy simultaneously. These negative results provide valuable theoretical boundaries.

- Novel distributed algorithm with the adaptive regularization which balances privacy noise and update stability, allowing performance to improve with scale.

- Asymptotic trade-off analysis: Theorem 7.1 shows how both exploitability and privacy budget can converge to zero as number of players $N \rightarrow \infty$ a clear improvement over prior results in aggregative and network games.

**Weaknesses:**

- Although the introduction cites “security games” and “financial markets” as motivating examples, the setting of the paper is quite abstract and conceptual. It is hard to imagine that actual scenarios in security or finance would actually fit into this framework, though I understand that the results give the theoretical limits in private fuly distributed games.

- Limited privacy scope: The paper only considers the edge-DP definition considered by [Huang et al. (2015)](https://arxiv.org/pdf/1401.2596) (edge-DP).

- Some central references seem to be missing: e.g. [Cyffers et al., 2022](https://proceedings.neurips.cc/paper_files/paper/2022/file/65d32185f73cbf4535449a792c63926f-Paper-Conference.pdf) is missing, as well as at other works by Cyffers and Bellet on decentralized ML. A recent paper by [El Mrini et al., 2024](https://proceedings.mlr.press/v235/mrini24a.html) also considers a local view of decentralized ML algorithms. [Cyffers et al., 2022](https://proceedings.neurips.cc/paper_files/paper/2022/file/65d32185f73cbf4535449a792c63926f-Paper-Conference.pdf) also considers a scenario where the adversary has only a local view and improve upon LDP guarantees. While the DP definitions are different: node-value level DP, meaning that the graph and edges stay constant instead of edge-DP like in this paper, I think thery are relevant.

- The fact that Algorithm 1 requires the harmonic mean of players degrees, $\bar{\mathcal{N}}$ and total number of players, i.e., global information, makes it less practical since I think some central entity would be required to communicate that information.

**Questions:**

- Why to only consider the edge-DP definition? Why not e.g. player-wise DP or DP for the values of the nodes?

- Could you think of practical scenarios where the results of this paper would be useful?

- Or: could you think of extending the results of this paper to a practical scenario where they would be useful?

- In Thm. 6.1, what is the improvement over LDP scenario, i.e., when everything is observed? It seems as if the guarantees of Thm. 6.1 get weaker as $N$ grows, or am I reading it wrong? So the vanishing error of Thm. 7.1 comes somehow from the averaging effect?

---

> ### Author Response · Authors · 2025-11-20
> **Response to Reviewer UsT1**
>
> We thank the reviewer for the positive assessment of our contribution, particularly regarding the rigorous formalization of edge-DP and the insightfulness of our impossibility results. We address the specific concerns and questions below.
>
> ## Missing References (Cyffers et al., 2022; El Mrini et al., 2024)
>
> > "Some central references seem to be missing: ... I think thery are relevant."
>
> Thanks! We have updated the related work to cite the papers on decentralized ML suggested, and discuss how their ideas relate to our (somewhat different) setting.
>
> ## Edge DP
>
> > "Why to only consider the edge-DP definition? Why not e.g. player-wise DP or DP for the values of the nodes?"
>
> Node differential privacy also makes sense in our setting. Compared to edge DP, node DP is a stronger notion, and the current technique in this paper may not extend to node DP in dense graphs. This is because adjacent games in terms of node DP may have $\Theta(N)$ different utility matrices, and it may affect the adversary's observations a lot. However, we remark that if the goal is to ensure the privacy of players' utility matrices, our paper shows that edge DP is sufficient.
>
> ## Practical Scenarios
>
> > "Could you think of practical scenarios where the results of this paper would be useful?"
>
> We acknowledge that the initial motivating examples were abstract. To address this:
>
> - **Market Example:** Consider a decentralized bilateral trading network. Players are traders, and actions represent pricing strategies. The graph edges represent trade relationships. Players wish to reach a market equilibrium to maximize profit, but their specific utility matrices (which encode sensitive reservation prices or private valuations) must remain private to prevent exploitation in future bargaining.
> - **New Experiments:** In the revised paper, we have added a simulation of such a market scenario. The results demonstrate that our algorithm enables traders to converge to near-equilibrium prices without revealing their private valuation functions to neighbors.
>
> The additional example has also been added to the introduction.
>
> ##  Practicality of Algorithm 1 (Global Information)
>
> > "The fact that Algorithm 1 requires the harmonic mean of players degrees, $\bar{\mathcal{N}}$ and total number of players, i.e., global information, makes it less practical since I think some central entity would be required to communicate that information."
>
> You correctly noted that Algorithm 1 uses global parameters (total players $N$ and harmonic mean of degrees). While true, this is a standard assumption in many distributed optimization frameworks. For instance, [1] also requires a global Lipschitz constant $C_2$ and synchronization of the learning rate across all players.
>
> In practice, these scalars can be estimated with high accuracy using simple decentralized consensus protocols (*e.g.*, average consensus) running for a few iterations prior to the game. This introduces negligible communication overhead compared to the game dynamics.
>
> ## Theorem 6.1
>
> > "In Thm. 6.1, what is the improvement over LDP scenario, i.e., when everything is observed? It seems as if the guarantees of Thm. 6.1 get weaker as $N$ grows, or am I reading it wrong? So the vanishing error of Thm. 7.1 comes somehow from the averaging effect?"
>
> Both $\clubsuit$ and $\spadesuit$ vanish as $N$ grows.
> - For dense graphs, $\clubsuit$ vanishes because $\log N \ll \bar{\mathcal{N}}$.
> - For sparse graphs, $\spadesuit$ vanishes since most players are far from $v_1$ and $v_2$.
>
> [1] Huang, Zhenqi, Sayan Mitra, and Nitin Vaidya. "Differentially private distributed optimization." Proceedings of the 16th International Conference on Distributed Computing and Networking. 2015.

---

### Official Review · Reviewer_ssdW · 2025-11-01

**Soundness:** 4
**Presentation:** 3
**Contribution:** 4
**Rating:** 8
**Confidence:** 3

**Summary:**

In this work, the authors establish both impossibility results and positive results in the problem of computing equilibria in polymatrix games.
Paper demonstrates that achieving both high accuracy and low privacy budget is impossible if the adversary has access to all communication channels or if Euclidean distance is used as the accuracy metric. Conversely, the authors propose an algorithm that achieves differential privacy while converging to coarse correlated equilibrium (CCE) with guarantees on exploitability. A key contribution is demonstrating that both privacy and accuracy can improve as the number of players increases, particularly for dense or sparse graph structures, through careful regularization and noise injection.

**Strengths:**

**Novel problem domain.** The work pioneers DP equilibrium computation in polymatrix games, a richer class than previously studied DP games such as "Two-Player Zero-Sum" games and "Aggregative" games. The notion of adjacent games that is used --- reasonable and simplifies the formal statements.

**Impossibility results (Lemmas 3.1 and 3.2).** Prior DP game-theory work mostly considered aggregative games or centralized settings, but showing a lower bound on the Nash-gap for DP algorithms under full eavesdropping is a strong negative result, highlighting a fundamental privacy–accuracy trade-off.

**Positive results (Th. 5.1–7.1).** Give us that both exploitability and privacy cost vanish as $N\to\infty$, this results is stated formally in Th. 7.1. The authors highlight differences with prior works, which suffer a differential privacy budget, no matter how large $N$ is, and vice versa.

**Method.** The proposed algorithm (_Find CCE in polymatrix games with DP guarantee_) achieve both perfect accuracy and DP guarantees as the number of players in the game increases. Also, the efficient implementation on PyTorch is provided.

**Weaknesses:**

**One limitation regarding the adversary model.** The positive results assume the adversary can only see a constant number of communication channels. In practice, an adversary might intercept many edges. If it sees $k$ channels, then the privacy cost scales by $k$. It would be helpful to discuss how realistic your “bounded channels” assumption is, and how performance degrades as $k$ grows.

**You can extend the limitations a bit.** As mentioned in limitations, you only discuss polymatrix games. But I would clarify that your impossibility results focus on zero-sum polymatrix games; it is not clear if similar impossibility holds for general-sum cases.

**Extended experiments.** I think you can add more problems to your codebase, e.g., different topologies of the polymatrix game’s graph.

**Questions:**

**CCE vs Nash.**
The algorithm is analyzed in terms of coarse correlated equilibrium (CCE) regret. Do the authors have comments on whether a Nash equilibrium (or approximate NE) can be learned with DP under similar conditions? Could the algorithm be modified to target NE?

**Graph assumptions.** Are there any assumptions on the polymatrix game’s graph? Since the privacy bound involves shortest path distances: could highly irregular graphs (e.g., diameter ~ log N vs diameter ~ N) affect the results then?

**Parameter choice.** Th. 7.1 requires setting $\eta,T,\sigma$ as functions of $N$. How sensitive are the results to these choices? In practice, how would one tune these parameters?

**Details Of Ethics Concerns:**

No concerns

---

> ### Author Response · Authors · 2025-11-20
> **Response to Reviewer ssdW**
>
> We thank the reviewer for the encouraging feedback, specifically for recognizing the novelty of the problem domain and the significance of our impossibility and positive results.
>
> ## Adversary bounded channel
>
> > "The positive results assume the adversary can only see a constant number of communication channels ... how performance degrades as $k$ grows."
>
> Since we consider decentralized equilibrium computation, our model assumes a largely reliable communication network where only a subset of channels is accessible to the adversary. We specifically address this case. We have added a discussion in the latest version at the beginning of Section 6.
>
> ## Lower bounds for general-sum polymatrix games
>
> > "As mentioned in limitations, you only discuss polymatrix games. But I would clarify that your impossibility results focus on zero-sum polymatrix games; it is not clear if a similar impossibility holds for general-sum cases."
>
> We have clarified this in Section 3 of the latest version. Zero-sum polymatrix games are a strict subclass of general-sum polymatrix games (specifically, the subset where total utility is constrained to zero). Consequently, any lower bound or impossibility result established for the zero-sum case is logically stronger and automatically applies to the broader general-sum case. If equilibrium computation is hard for the specific zero-sum case, it is inherently hard for the general case.
>
> ## Extended Experiments
>
> > "I think you can add more problems to your codebase, e.g., different topologies of the polymatrix game’s graph."
>
> Thank you for this suggestion. We have expanded the experimental section and codebase to include two additional polymatrix game topologies:
> - Cluster Polymatrix Game: Players are grouped into clusters with dense intra-cluster connections and sparse inter-cluster connections. This models scenarios where players form distinct communities with limited external interaction / exhibit strong local modularity.
> - Locality Game: This models players on a 2-D plane where interactions occur only between a player and their $K$-nearest neighbors. This structure mimics transaction networks where agents only trade with local neighbors.
>
> ## CCE vs. Nash Equilibrium
>
> > "Do the authors have comments on whether a Nash equilibrium (or approximate NE) can be learned with DP under similar conditions? Could the algorithm be modified to target NE?"
>
> In the general-sum setting, computing a Nash Equilibrium (NE) is known to be PPAD-complete [1], making it computationally intractable for efficient learning algorithms. However, for the specific subclass of zero-sum polymatrix games, our algorithm does converge to the NE (by taking the marginal probability distribution of each player), which is consistent with existing literature [2]. Therefore, while the algorithm targets CCE in the general case due to complexity constraints, it successfully recovers NE in the zero-sum case.
>
> ## Graph Diameter
>
> > "Since the privacy bound involves shortest path distances: could highly irregular graphs (e.g., diameter ~ log N vs diameter ~ N) affect the results then?"
>
> Yes, the graph diameter will affect the result. The results of the sparse graph can be generalized to graphs with a large diameter, such as the transportation network.
>
> ## Sensitivity
>
> >  "Th. 7.1 requires setting $\eta,T,\sigma$ as functions of $N$. How sensitive are the results to these choices?"
>
> In our experimental setup, we fix the noise scale $\sigma = \frac{1}{\sqrt{T}}$. This leaves only the learning rate $\eta$ and the time horizon $T$ to be tuned. In practice, one can select $\eta$ and $T$ to satisfy a specific differential privacy budget. Furthermore, because the algorithm is grounded in no-regret learning, it is robust to parameter selection: variations in $\eta$ and $T$ primarily affect the rate of convergence, but the algorithm remains guaranteed to converge.
>
> [1] Daskalakis, Constantinos, Paul W. Goldberg, and Christos H. Papadimitriou. "The complexity of computing a Nash equilibrium." Communications of the ACM 52.2 (2009): 89-97.
>
> [2] Cai, Yang, et al. "Zero-sum polymatrix games: A generalization of minmax." Mathematics of Operations Research 41.2 (2016): 648-655.

---

> > ### Comment · Reviewer_ssdW · 2025-11-20
> > **did you actually update the manuscript and the supplementary material?**
> >
> > I thank the Authors for the provided feedback.
> > I am checking the improved manuscript and their code. If it is possible, I would like the Authors to highlight the adjusted text, e.g., colour it somehow. I also do not see modifications in the code at this point.
> >
> > I am waiting for authors to clarify wether they updated the entire submission.
> > Meanwhile, I am satisfied with the clarifications given and will vote for acceptance.

---

> > > ### Author Response · Authors · 2025-11-20
> > >
> > > Thank you so much for your prompt reply. The manuscript and the code are both updated.

---

### Official Review · Reviewer_8A83 · 2025-11-01

**Soundness:** 3
**Presentation:** 3
**Contribution:** 3
**Rating:** 6
**Confidence:** 2

**Summary:**

This paper studies the problem of zero-sum polymatrix games under differential privacy (DP). The notion of DP is defined over adjacent games that differ by a single utility matrix.

Building on this definition, the authors establish two lower-bound results, showing that to achieve meaningful privacy and convergence guarantees, (i) the adversary must not have access to all communication channels, and (ii) the accuracy metric cannot be based on the Euclidean distance. The paper further provides upper-bound results demonstrating that low privacy parameters and small exploitability can be achieved when the adversary has access to only a single user’s communication channel.

**Strengths:**

The lower-bound results are solid contributions, as they establish the necessary conditions for ensuring valid privacy and utility guarantees. I also appreciate that the authors provide detailed explanations and intuitions behind most of the theoretical results, which greatly help readers better understand these concepts.

**Weaknesses:**

1. In Theorem 6.1, the privacy of a single user’s observation is defined as the average privacy loss across all users. However, the corresponding privacy notion in Equation (7) is defined as a maximum over all users, i.e., it bounds the worst-case privacy loss. Therefore, the comparison between the lower- and upper-bound results may not be entirely fair, since Equation (7) imposes a stronger constraint on privacy loss (maximum vs. average). It would be helpful to clarify whether Theorem 6.1 still holds if we instead upper bound the maximum privacy loss rather than the average.
2. Following Lemma 3.2 (Lines 279–281), the authors present an example illustrating the instability of the best response with respect to other players’ strategies. I am curious how this instability is circumvented when exploitability is used as the performance metric instead of Euclidean distance.
3. The dependence on $N$ for both the accuracy and privacy loss in the upper-bound results appears quite weak, scaling with $O((\log N)^{-1/3})$. The authors note that this is the first result showing diminishing accuracy with sufficiently large $N$, which is an interesting finding. It would strengthen the discussion to also include a comparison with the non-private convergence rate, to better illustrate the performance gap between private and non-private settings.

**Questions:**

All questions are included in "Weaknesses" section.

---

> ### Author Response · Authors · 2025-11-20
> **Response to Reviewer 8A83**
>
> We thank the reviewer for their constructive feedback and for recognizing the soundness of our lower-bound results and the clarity of our theoretical intuition. Below, we address the specific questions raised in the "Weaknesses" section.
>
> ## Consistency of Privacy Definitions (Theorem 6.1 vs. Equation 7)
>
> > "In Theorem 6.1, the privacy of a single user’s observation is defined as the average privacy loss across all users. However, the corresponding privacy notion in Equation (7) is defined as a maximum over all users, i.e., it bounds the worst-case privacy loss."
>
> We clarify that the comparison between our lower and upper bounds is indeed fair. Equation (7) corresponds to the lower bound established in Lemma 3.2. In this lemma, we bound the average privacy budget, $\frac{1}{N}\sum_{i=1}^N \epsilon_i$, rather than the maximum privacy loss. Consequently, both the lower bound (Lemma 3.2) and the upper bound (Theorem 6.1) are analyzed under the average privacy constraint. To prevent future confusion, we have revised the text following Equation (7) in the updated manuscript to explicitly state that the definition relies on the average budget.
>
> ## Instability and the Exploitability Metric
>
> > "I am curious how this instability is circumvented when exploitability is used as the performance metric instead of Euclidean distance."
>
> The reviewer correctly notes that best-response dynamics can be unstable in Euclidean space. This instability arises because the argmax operator (best response) is discontinuous with respect to other players' strategies. In contrast, the exploitability function is Lipschitz continuous, allowing us to circumvent these stability issues.
>
> To illustrate this difference, consider the example under Lemma 3.2 where the row player shifts their strategy slightly from $(0.51, 0.49)$ to $(0.49, 0.51)$ while the column player remains fixed at $(0, 1)$. Under the Euclidean metric, the best response jumps discontinuously from $(0, 1)$ to $(1, 0)$. However, the exploitability only changes from $0.0$ to
>
> $$(-0.49) - (-0.51) = 0.02$$
>
> This variation is significantly smaller than the jump in the best response. By utilizing exploitability, we leverage this continuity to provide convergence guarantees that would be unattainable using the Euclidean distance metric.
>
> ## Dependence on $N$ and Convergence Rates
>
> > "It would strengthen the discussion to also include a comparison with the non-private convergence rate, to better illustrate the performance gap between private and non-private settings."
>
> We agree that a comparison strengthens the discussion, and we have added it in the latest version at the end of Section 7. In the non-private setting, the convergence rate is $O(\frac{1}{\eta T} + \eta)$. Our private algorithm adds overhead primarily due to DP noise and regularization, but the convergence remains efficient.
>
> To clarify the dependence on $N$ (where $\bar {\mathcal{N}}=O(N^p)$): the value $O(\frac{1}{(\log N)^{1/3}})$ or $O(N^{-4p/27})$ are the error bound at a specific time step, not the rate itself. We achieve:
>
> - $\tilde O(N^{-4p/27})$ error when $T=\tilde O(N^{8p/9})$.
> - $O(\frac{1}{(\log N)^{1/3}})$ error when $T=O(\log N)$.
>
> This confirms that while accuracy diminishes slightly with sufficiently large $N$ to satisfy privacy, the algorithm still converges rapidly.

---

### Meta-Review · Area_Chair_V5Lm · 2026-01-06

**Summary:**

Three reviewers give positive rating and only one reviewer gives a 4. The authors have addressed the negative comments such as whether the utility matrices are known, the motivation for privacy preservation, comparison with bandit feedback, etc.

**Reviewer Concerns:**

Most comments have been addressed.

**Reviewer Scores:**

Unchanged or the negative reviewer (rating 4) may improve his/her score

---

### Decision · Program_Chairs · 2026-01-26

Accept (Poster)